# Structural basis of organic cation transporter-3 inhibition

Basavraj Khanppnavar[1,2,9], Julian Maier [3,9], Freja Herborg [4], Ralph Gradisch [3], Erika Lazzarin[3], Dino Luethi [3], Jae-Won Yang [3], Chao Qi [1], Marion Holy[3], Kathrin Jäntsch[3], Oliver Kudlacek[3], Klaus Schicker[3], Thomas Werge [5,6,7,8], Ulrik Gether [4], Thomas Stockner [3]✉, Volodymyr M. Korkhov [1,2]✉ & Harald H. Sitte [3]✉

Organic cation transporters (OCTs) facilitate the translocation of catecholamines, drugs and xenobiotics across the plasma membrane in various tissues throughout the human body. OCT3 plays a key role in low-affinity, high-capacity uptake of monoamines in most tissues including heart, brain and liver. Its deregulation plays a role in diseases. Despite its importance, the structural basis of OCT3 function and its inhibition has remained enigmatic. Here we describe the cryo-EM structure of human OCT3 at 3.2 Å resolution. Structures of OCT3 bound to two inhibitors, corticosterone and decynium-22, define the ligand binding pocket and reveal common features of major facilitator transporter inhibitors. In addition, we relate the functional characteristics of an extensive collection of previously uncharacterized human genetic variants to structural features, thereby providing a basis for understanding the impact of OCT3 polymorphisms.

Organic cation transporters (OCTs; Fig. 1a) are low-affinity, high-capacity transporters[1] which act complementary to high-affinity, low-capacity neurotransmitter:sodium symporters (solute carrier family 6, SLC6)[2] in regulating and maintaining the extracellular equilibrium of monoamine neurotransmitters[3]. Norepinephrine, dopamine and serotonin belong to this family; interaction with their cognate receptors and subsequent signaling events are essential for physiological function and play multiple roles in a variety of pathologies[4,5]. In addition, OCTs are responsible for cellular uptake of cationic drugs in various tissues[1].

Organic cation transporter 3 (OCT3, SLC22A3) was identified as a corticosterone-sensitive catecholamine transporter in 1998 and belongs to the SLC22 family of the major facilitator superfamily (MFS)

of transporters[6]. OCT3 is more widely expressed than OCT1 and OCT2, which are primarily located in liver and kidney, respectively[7]. Due to the effect of OCTs on the pharmacokinetic fate of therapeutically relevant drugs, FDA and EMA recommend to screen compounds for possible interaction with OCTs[8]. A number of single nucleotide polymorphisms (genetic variants) have been identified in OCT3; in some instances the missense mutations have been linked to specific functional properties of the transporter and associations identified between genetic variants and cardiovascular disease[9,10], type 2 diabetes[11,12] and cancer[13].

Recent findings reinforce the role of OCT3 in various (patho-)physiological processes. Norepinephrine uptake in cardiomyocytes is necessary for cardiac contractility, which is mainly mediated by

[1]Laboratory of Biomolecular Research, Paul Scherrer Institute, Villigen, Switzerland. [2]Institute of Molecular Biology and Biophysics, ETH Zurich, Zurich, Switzerland. [3]Institute of Pharmacology, Center for Physiology and Pharmacology, Medical University of Vienna, Vienna, Austria. [4]Department of Neuroscience, Faculty of Health and Medical Sciences, University of Copenhagen, Copenhagen, Denmark. [5]Institute of Biological Psychiatry, Mental Health Services Copenhagen, Copenhagen, Denmark. [6]Department of Clinical Medicine, University of Copenhagen, Copenhagen, Denmark. [7]The Lundbeck Foundation Initiative for Integrative Psychiatric Research (iPSYCH), Aarhus, Denmark. [8]The Globe Institute, Lundbeck Centre for Geogenetics, University of Copenhagen, Copenhagen, Denmark. [9]These authors contributed equally: Basavraj Khanppnavar, Julian Maier. ✉e-mail: thomas.stockner@meduniwien.ac.at; volodymyr.korkhov@psi.ch; harald.sitte@meduniwien.ac.at

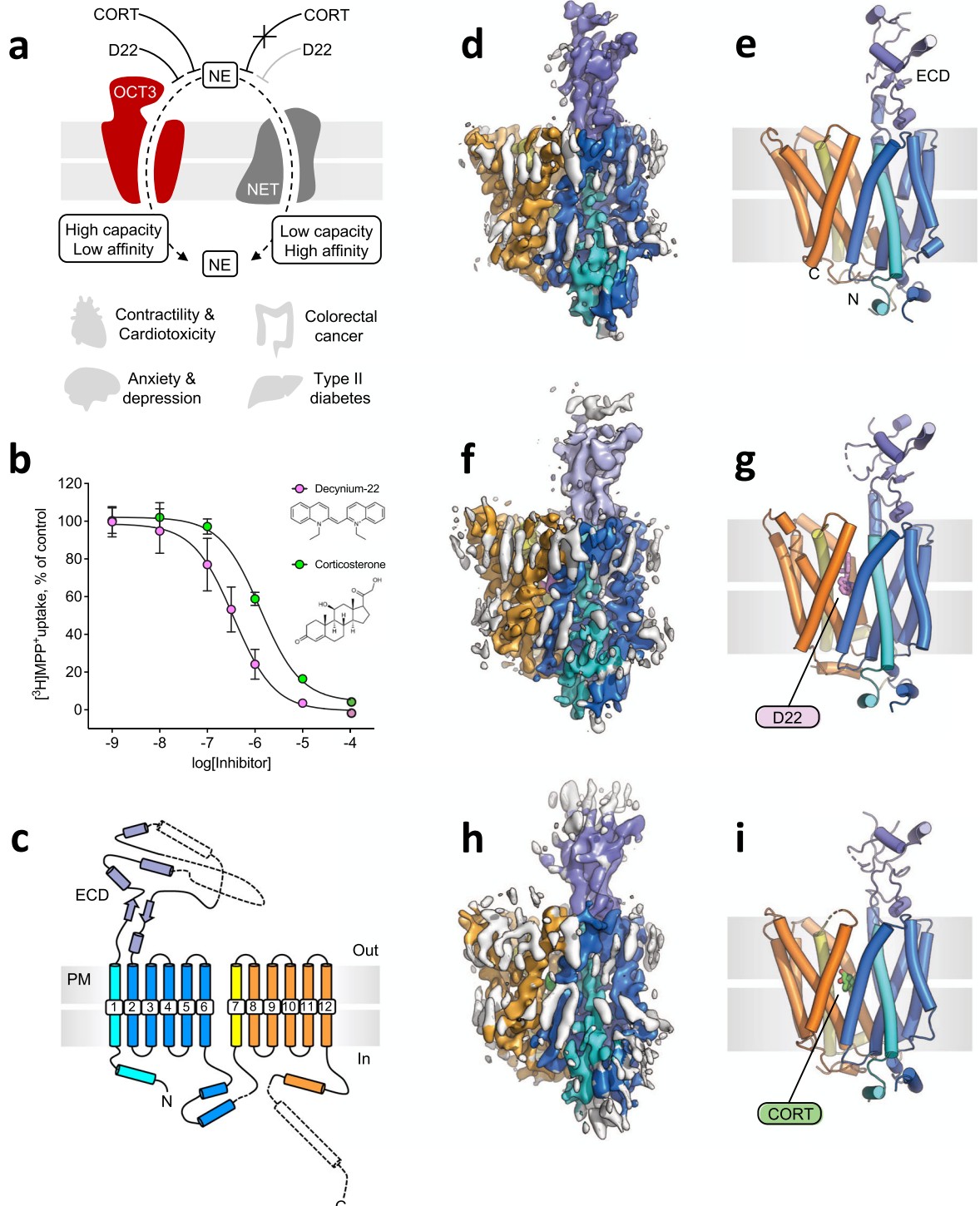

**Fig. 1 | Structure and function of OCT3. a** Schematic representation of OCT3; key features of the transporter are illustrated in the panel. **b** OCT3 transport inhibition by decynium-22 (D22) and corticosterone (CORT). The values correspond to mean ± SD; $n = 3$ represents three biologically independent experiments that are conducted with three technical repeats. **c** A scheme depicting the topology and the secondary structure elements of OCT3. **d**, **e** The cryo-EM map (**d**) and model of OCT3[14]. OCT3 in nanodiscs at 3.2 Å resolution. The colors of the protein correspond to those in c; annular lipids are colored gray. **f**, **g** Same as **d**, **e**, for OCT3-D22 complex at 3.6 Å resolution (D22 colored violet). **h**, **i** Same as **f**, **g**, for OCT3-CORT complex at 3.7 Å resolution (CORT colored green). Source Data are available as a Source Data file.

OCT3[14]. OCT3 also mediates the uptake of doxorubicin into the myocardium and thus contributes to its dose-limiting toxicity[15]. OCT3 polymorphisms and reduced hepatic OCT3 expression, caused by cholestasis and liver fibrosis, lead to reduced metformin uptake, thereby impacting its therapeutic action through changed pharmacokinetics[7,16,17]. In addition, loss of OCT3 is associated with progression of liver fibrosis[18] and hepatocellular carcinoma[19]. Colorectal cancer cells overexpress OCT3 which renders them susceptible to oxaliplatin-induced cytotoxicity[20]. Finally, inhibition of OCT3 allows for raising extracellular levels of serotonin and norepinephrine in the

brain and may thus present an alternative to currently approved antidepressants in the treatment of depressive disorders[21,22].

Potent and selective inhibition of OCT3 may have immediate medical implications. However, specific ligands are scarce and progress is hampered by the lack of structural information. Prototypical ligands of OCT3 are decynium-22 (D22) and the OCT3-selective steroid hormone corticosterone (CORT). We set out to determine the structure of OCT3 alone, as well as in complexes with its two characteristic inhibitors. In addition, we delineated the structure-function relationship of OCT3 by examining how genetic variants of OCT3, which were discovered by exome-sequencing in a cohort of 17,339 subjects, affect transport activity.

## Results

### Structure determination of human OCT3

We first set out to determine the structure of OCT3 using cryo-EM and single particle analysis. We purified and reconstituted OCT3 into nanodiscs (MSP1D1 filled with brain polar lipids; Supplementary Fig. 1a-b) and subjected the sample to extensive cryo-EM data collection and image processing (Supplementary Fig. 1c and 2, Supplementary Table 1). We obtained a 3D reconstruction of OCT3 (apo-state) at 3.2 Å resolution (Fig. 1d-e, Supplementary Fig. 2, 5a-b and 6) and completed the model by combining the experimentally determined structure with an in silico model generated by AlphaFold (detailed in Materials and Methods, Supplementary Fig. 6 and 7f).

The structure revealed a classical MFS fold for OCT3, with the transporter composed of twelve TM helices (TM1-12) in an outward-facing conformation. The translocation pathway is located at the interface of the two 2-fold pseudo-symmetrically-related transmembrane domains consisting of TM1-6 and TM7-12. The substrate binding site is located in the center of the transporter between the two domains, halfway through the membrane. The structure features a prominent, partially resolved density of the extended extracellular loop 1 or ectodomain (ECD), which is adjacent to the outward-open translocation pathway (Fig. 1c-d). Upon expression in human embryonic kidney-293 cells, human OCT3 transported its substrate MPP$^+$ (1-methyl-4-phenylpyridinium) and showed inhibition by two key molecules, D22 and CORT (Fig. 1b).

### Structures of D22- and corticosterone-bound OCT3

We incubated purified and reconstituted OCT3 with saturating concentrations of D22 and CORT (1 mM, Fig. 1b). We subsequently determined the structures of OCT3 in D22- and in CORT-bound states at 3.6 Å and 3.7 Å resolution, respectively (Fig. 1f-i, Supplementary Fig. 3, 4 and 5c-f). A comparison of the two inhibitor-bound states with the apo-state of OCT3 clearly showed that the compounds are readily accommodated by the protein in its outward-facing conformation with minimal conformational changes (Fig. 2, Supplementary Fig. 8). The root mean square deviation (RMSD) between all atoms of the apo-state and each of the inhibitor-bound states is 0.01 Å (apo vs D22) and 0.69 Å (apo vs CORT). CORT placement was assisted by molecular dynamics (MD) simulations, as detailed in "Materials and Methods" (Supplementary Fig. 7, 8 and 9). The cryo-EM structures of OCT3 and the full-length models of the loops, completed with AlphaFold, were further validated using MD simulations. The trajectories showed stable structures for apo-OCT3 as well as for the D22- and CORT-bound structures (Supplementary Fig. 7 and 9). Addition of the AlphaFold-based missing parts of the extracellular and intracellular domains increased the stability of OCT3 secondary structure and reduced the deviation from the starting structures (lower RMSD and root mean square fluctuation (RMSF), Supplementary Fig. 7).

Both compounds occupy the substrate translocation pathway resulting in its steric blockage. Interestingly, the binding sites for D22 and CORT partially overlap, but the orientations of the compounds within the pocket are distinct. The cationic D22 is bound to OCT3 in a pose oriented perpendicular to the membrane plane (Fig. 2d). D22 binds to only one side of the outer vestibule, making several interactions with TM7 and TM11 (Fig. 2g). In contrast to D22, CORT is positioned roughly parallel to the plane of the lipid bilayer (Fig. 2f) and makes more extensive contacts within the binding pocket, with 11 residues (Fig. 2h).

Although a number of residues are shared between the binding sites of CORT and D22, the mode of binding of these two compounds differs substantially (Fig. 2c-h). Furthermore, in the presence of CORT, the OCT3 density map features several additional elements located at the entrance to the outward-open binding pocket (Supplementary Fig. 10a). We interpret this as a low-affinity site that may accept weakly bound CORT molecules in alternate poses (Supplementary Fig. 10b). The poses of D22 and CORT suggest that the mechanism of transport inhibition by these two compounds may differ. The tightly bound D22, positioned perpendicular to the membrane plane within the binding site, may prevent outward to inward conformational changes in OCT3. In contrast, CORT occupies a larger footprint within the binding site. It may also prevent outward-to-inward rearrangements of the transporter by occupying multiple non-specific binding sites and thus "clogging" the translocation pathway.

### Molecular basis of OCT3 ligand specificity

CORT is selective for OCT3[6,23,24]. In contrast, D22 binds to and inhibits all OCTs (Fig. 3e), showing only slightly higher affinity for OCT3. However, residues interacting with D22 and CORT in OCT3 are largely conserved among the three OCTs (Fig. 3f). The ligand binding sites differ in only six residues: F36 (TM1), F250 (TM5), I254 (TM5), F450 (TM12), E451 (TM12) and Y454 (TM12). These six residues correspond to C36, F244, L248, I446, Q447, C450 in OCT1 and to Y37, Y245, L249, Y447, E448, C451 in OCT2 (Fig. 3f). We compared the OCT3 structures to the homology models of OCT1 and OCT2 generated by AlphaFold[25] (Fig. 3g). The OCT1 substrate binding pocket lacks negatively charged residues, compared to OCT2 and 3. The tyrosine residues in OCT2 (Y245, Y447) that correspond to F250 and F450 in OCT3 increase the hydrophilicity of the substrate binding pocket and provide two additional hydrogen bond donors.

### Negatively charged residues line the translocation pathway

The movement of organic cations across the plasma membrane earned OCTs their name[26]. The prerequisite for efficient cation transport is the presence of a regulated anionic surface that attracts the cationic ligand on one side of the membrane, induces a conformational change of the transporter and leads to the release of the compound on the other side of the membrane in one productive transport cycle. The structure of OCT3 features various negatively charged residues lining the substrate translocation pathway: D155, E232, D382, E390, E451, E459 and D478. Several of these residues are in proximity of the OCT3-bound D22 and CORT molecules (Supplementary Fig. 11a). A comparison of the OCT3 structure with the homology models of OCT1 and OCT2, the two closest homologs of OCT3 in the SLC22 family, as well as with the organic anion transporters (OATs, also members of SLC22 family of transporters) illustrates the negative and positive electrostatic potential in OCT and OAT translocation pathways, respectively (Supplementary Fig. 11). The electrostatic properties of these transporters appear to be consistent with their functional annotation as anion or cation transporters. The differences in charge distribution within the substrate binding pockets of OCTs and OATs are consistent with their substrate preference (Supplementary Fig 11).

### Comparison of OCT3 ligand binding to other MFS transporters

Despite the availability of several MFS transporter structures, only few have been determined in states bound to their small molecule substrates or to inhibitors[27–32]. A comparison of OCT3-bound D22 and OCT3-bound CORT to other small molecule-bound transporters

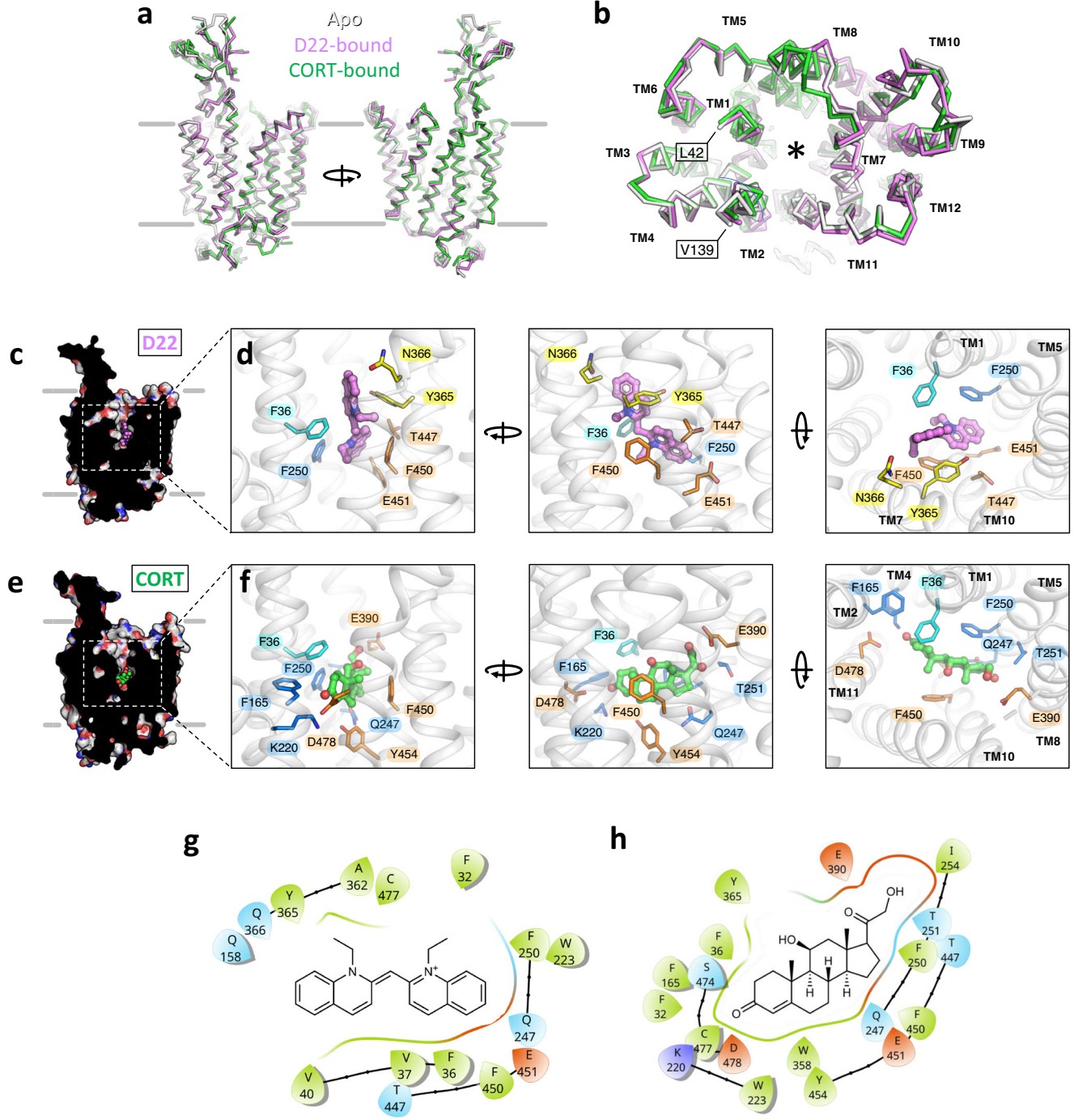

**Fig. 2 | Comparison of the apo-state and the ligand-bound states of OCT3.**
**a** Structural alignment of the three indicated states of OCT3 shows an overall high degree of similarity. **b** Same as in a, view from the extracellular space. The positions of each transmembrane (TM) helix are indicated. The residues flanking the extended extracellular loop 1 (ECL1) are indicated with boxes (L42 and V139). The translocation pathway / ligand binding site is indicated by an asterisk. **c** A sliced view of D22, showing the inhibitor buried deep in the substrate translocation pathway. **d** The expanded views of the D22 binding site in different orientations (*left* the same orientation as the one in **c**), indicating the residues within 4 Å distance of the inhibitor. **e**, **f** Same as **c**, **d**, for OCT3-CORT. The TM domains containing the residues in close proximity to the ligand are indicated in the right-most panel (**d** and **f**). **g**, **h** 2D interaction plot showing the residues interacting with D22 and CORT.

(Fig. 3a-d) captured in an outward-facing state shows that the inhibitors of these transporters utilize a common, simple and effective mechanism of inhibition: binding to the substrate binding site in connection with inhibition of conformational changes essential for substrate translocation. The region of the transporter involved in these inhibitory interactions involves residues in TM1, TM2, TM5, TM7, TM8 and TM11, which are exposed to the substrate translocation path (Fig. 3a-d). As noted above, in OCT3 the orientation of D22

perpendicular to the membrane plane may not only block the translocation pathway, but may also prevent an outward-to-inward rearrangement of the OCT3 conformation. It is possible that the effective inhibitors of the other MFS transporters share this feature with D22, occupying a position that blocks the translocation pathway and constrains the conformational flexibility of the protein. This may have important implications for rational design of MFS transporter inhibitors.

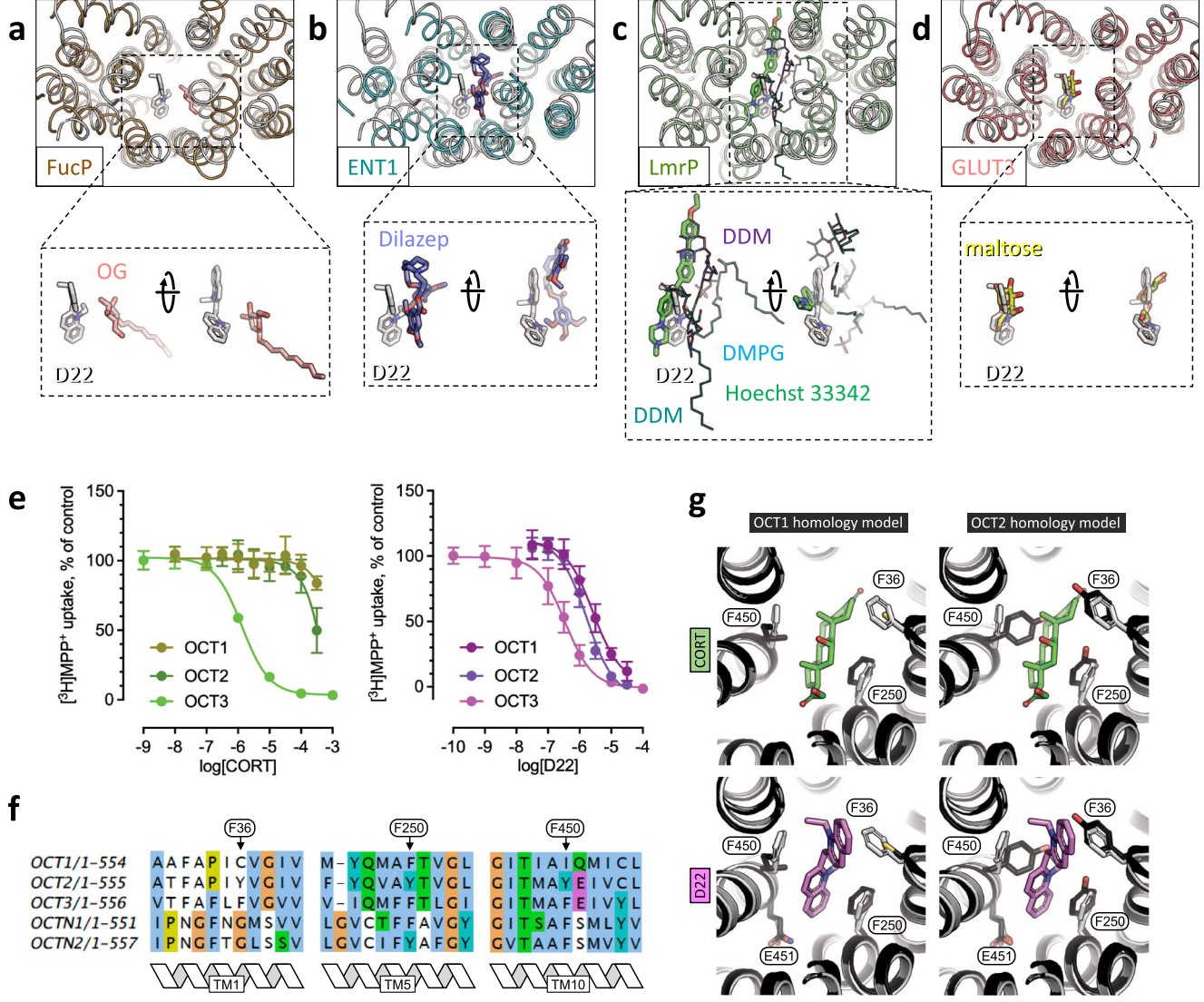

**Fig. 3 | Molecular basis of OCT3 ligand specificity. a–d** Comparison between OCT3-D22 and four different MFS transporter structures in outward-facing, ligand bound states, including FucP (PDB ID: 3o7q), ENT1 (PDB ID: 6ob7), LmrP (PDB ID: 6t1z) and GLUT3 (PDB ID: 4zwc). The dotted lines/boxes show the zoomed in views of the isolated ligands (*left*, same orientation as in the main panel; *right* – rotated -90°). **e** Inhibition of OCT1, 2 and 3 transport by D22 and CORT (mean ± SD; $n = 3$).

$n = 3$ represent three biologically independent experiments for each cell line. **f** Sequence alignment of OCT1-3, OCTN1 and OCTN2, indicating the positions of the key OCT3 residues involved in ligand binding (and varied among the homologs): F36, F250 and F450. **g** Comparison of the CORT- and D22-bound states in the experimentally determined OCT3 structures (white) with the OCT1 and OCT2 homology models (black). Source Data are available as a Source Data file.

## Lateral access into the substrate translocation pathway

The outward-open state of the lipid-reconstituted OCT3 is surrounded by several lipid densities (Fig. 1d). The protein structure features a V-shaped lateral opening at the interface between the two halves of the protein ("V-site"; Supplementary Fig. 12). Several conserved lipid densities are present at this site, indicating the margins of the lipid bilayer (Supplementary Fig. 13). We presently cannot unambiguously assign the identity of the lipids in this region (brain polar lipids, which include cholesterol and phospholipids, were used for nanodisc reconstitution). This V-site may serve as an access pathway for hydrophobic molecules that diffuse into the OCT3 translocation pathway. Although similar features are present in other MFS transporters[27], the structure of OCT3 in a nanodisc allows us to visualize the lipid densities at this lateral opening (Supplementary Fig. 12).

## Genetic variants of OCT3

We investigated the occurrence of coding genetic variants in a large exome sequencing dataset from the iPSYCH2012 cohort[33]. This dataset includes 4885 healthy controls and 12454 patients diagnosed with at least one of five major psychiatric diseases: ADHD, autism-spectrum disorders, bipolar disorder, depression or schizophrenia. In total, we identified 58 different coding variants in 402 individuals in the combined cohort of cases and controls (Supplementary Table 4). These included 27 novel and 31 previously reported variants according to the Genome Aggregation Database database[34]. We then performed carrier-based association analyses to compare the carrier frequency of coding variants among control subjects and patients. Remarkably, we found a significant enrichment of coding variants in control subjects with a 1.29 fold higher carrier frequency (2.76% in controls vs. 2.14% in cases, $p = 0.0159$, OR = 0.771; 95% CI [0.624-0.949], Fisher's exact test, Supplementary Table 5), suggesting a potential protective effect of OCT3 coding variants against psychiatric disease. The combined group of coding variants encompasses potential 'loss of function' (LoF), non-functional, and potential gain of function variants of varying effect sizes. A separate carrier-based association analysis of the variants that completely disrupt OCT3 function, i.e. the identified nonsense,

frameshift and splice site variants revealed an even more pronounced enrichment in control subjects relative to cases, with carrier frequencies of 0.512% and 0.233% respectively ($p = 0.0057$, OR = 0.454; 95% CI [0.268-0.788], Fisher's exact test). For the remaining group of coding missense variants, we observed an interesting tendency for accumulation in control individuals ($p = 0.149$, OR = 0.846; 95% CI [0.846−1.062], Fisher's exact test, Supplementary Table 5). In light of these genetic data, we performed detailed functional investigations of 24 selected missense variants to interrogate the structure-function relationship of OCT3 (Fig. 4b, Supplementary Table 6).

### Structure-function analysis of OCT3 genetic variants

The herein investigated genetic variants were mapped on the structure of OCT3 (Fig. 4a) and functionally characterized in vitro (Fig. 4b, Supplementary Fig. 14-17, Supplementary Table 6). While some of the OCT3 mutants displayed protein expression or cell surface trafficking defects, many reached at least 60% of wild-type surface expression but showed reduced uptake function (Fig. 4b). A classification of the function-perturbing mutations suggests the existence of three classes: (a) mutations directly affecting the substrate translocation pathway (indicated by reduced uptake but conserved surface expression), (b) mutations modulating the conformational transitions (reducing uptake) (c) mutations affecting protein folding and cell surface expression (indicated by lower surface expression measured by confocal microscopy and higher Förster Resonance Energy Transfer (FRET) due to intracellular accumulation of proteins).

The structure of OCT3 allows for a rational interpretation of the large collection of functional data. This can be summarized as follows: (i) Genetic variants located in the ECD of OCT3 do not or barely affect uptake velocity and affinity for ligands; in fact, some variants even have higher $V_{max}$- and $K_m$-values than OCT3-WT (e.g. P54L, R120H). (ii) Genetic variants located in proximity of residue 340 interfere with folding and cause ER retention, with only no or little surface expression of D340G and R348W. (iii) Genetic variants located in the ICL (G235A, R298Q, with the exception of D340G) do not affect transport and ligand affinity. (iv) Genetic variants located in TM1, 5, 7, 9, 10, 11 and 12 mostly led to reduced transport velocity; (v) genetic variants near the translocation pathway (TM4, TM11) have pronounced effects on uptake and affinity (Fig. 4a, b, Supplementary Fig. 14−17, Supplementary Table 6).

Subsequently, we focused on mutations directly affecting substrate translocation and selected the six mutants in immediate proximity of the substrate binding site or the translocation pathway (Fig. 4a–c). Functional assays, including MPP$^+$ uptake and D22 and CORT uptake inhibition assays, revealed that three of the six selected mutants stand out: R212C, W223R and Y461H (Fig. 4d–f).

Substitution of W223 by arginine did not impair delivery of the protein to the cell surface but resulted in an inactive transporter (Supplementary Fig. 18). Residue W223 is central to the substrate binding site, making extensive interactions with CORT. Substitution of W223 by R removes the large aromatic sidechain and places a positive charge into the substrate binding site. This is likely to interfere with binding of organic cations by electrostatic repulsion. Simulations showed that the environment of W223, exemplified by the distances of W223 to Y227 and Q247, is strongly dependent on the presence of the ligands within the binding site (Fig. 5a, b). Thus, it is likely that W223R mutation does not only affect protein-substrate interactions, but also influences the neighboring residues within the substrate translocation pathway.

The mutants R212C and Y461H both displayed reduced $V_{max}$ and lower $K_m$ for transport of MPP$^+$ and showed higher affinity for the inhibitors D22 and CORT (Fig. 4d–f, Supplementary Table 3). These two residues are both located within the hydrophobic core but on opposite sides, i.e. R212 (TM4) in the vicinity of the extracellular vestibule and Y461 (TM10) close to the intracellular end of the substrate

permeation pathway (Fig. 4a, c) are sandwiched between TM1, TM2, TM3, TM4, and TM6 and between TM7, TM10 and TM11, respectively. Both residues are shielded from the substrate translocation path and may have a role in stabilizing TM4 (R212) and TM10 (Y461), which line the substrate translocation pathway.

Residue Y461 is located in a hydrophobic pocket and forms a very stable hydrogen bond with T351 on TM7 (Fig. 5c, d), thereby fixing the distance between TM7 and TM10. Binding of the inhibitors D22 and CORT appears to destabilize this hydrogen bond, based on MD simulations (Fig. 5c, d). The Y461H mutation is likely to break the hydrogen bond and through its much higher polarity attract water molecules. Together these changes may destabilize the central helix TM10. It is conceivable that Y461 plays a role in maintaining the structural integrity in the whole domain-based conformational transitions of OCT3 during the transport cycle.

The side chain of R212 is in proximity of the main-chain atoms of L35, V39 (TM1), T157 (TM2) and the side chains of T157 (TM2) and Q215 (TM4), and Q271 (TM6) (Fig. 5e, f). TM1, TM2 and TM4 contain residues involved in substrate/inhibitor interactions: mutation of R212 to a cysteine led to a reduction in transport $V_{max}$ and a decrease in $K_m$. This is consistent with the assumption that interactions with R212 are required for an effective transport cycle. Our MD simulations show that R212 participates in an extended hydrogen bonding network, which includes the nearby polar residues and 2-4 water molecules that fill the intra-membrane cavity surrounding R212 (Fig. 5e, f). In apo OCT3, this network is very dynamic, as R212 cannot establish all interactions simultaneously. By slightly shifting its position, R212 interacts directly with the side chains of T157, Q215 and Q271, while the additional interactions are formed through a water bridge. Binding of the inhibitors D22 or CORT induces small structural changes in OCT3, reducing the conformational dynamics of R212 as it partitions towards Q215 and Q271. The differential effects of the ligands on R212 environment and dynamics, combined with the functional studies of the R212 mutants, suggest an important role of this residue in controlling the conformational changes of OCT3 during ligand binding and translocation.

## Discussion

The SLC22 family comprises >30 transporters, which facilitate the transport of organic cations (OCTs), anions (OATs) and zwitterions (OCTNs)[1]. Collectively, these transporters define the pharmacokinetics of a vast array of drugs and xenobiotics[35,36]. Herein, we describe the cryo-EM structure of OCT3 and provide the first direct insights into the organization of a SLC22 member, its substrate permeation pathway and ligand binding pocket.

Both ligands of which we herein report cryo-EM structures, are handled by OCT3 in different ways which only partially overlap. It is not surprising, however, that the binding site of OCT3 allows accommodation of many diverse binding partners; the behavior rather substantiates the poly-specificity of a class of transporters which interact with a wide and complex array of compounds: from the antiviral drug abacavir and the antidiabetic drug metformin to the antineoplastic drug sunitinib[1].

However, the current consensus in the field is that both corticosterone and D22 are non-transported inhibitors with scarce evidence that D22 may accumulate into astrocytes via an OCT3 dependent mechanism[37]. It remains to be determined whether OCTs are able to move inhibitors across the membrane. This property would be similar to multidrug transporters, such as ABCB1 and ABCG2, which are capable of transporting a variety of organic compounds, including their inhibitors elacridar and tariquidar[38]. Our structures show that OCT3 inhibitors may sterically block the translocation pathway. Future studies will be necessary to determine whether these poses of the inhibitors completely impede transport, or whether the transporter is nevertheless capable of moving the inhibitor molecules across the

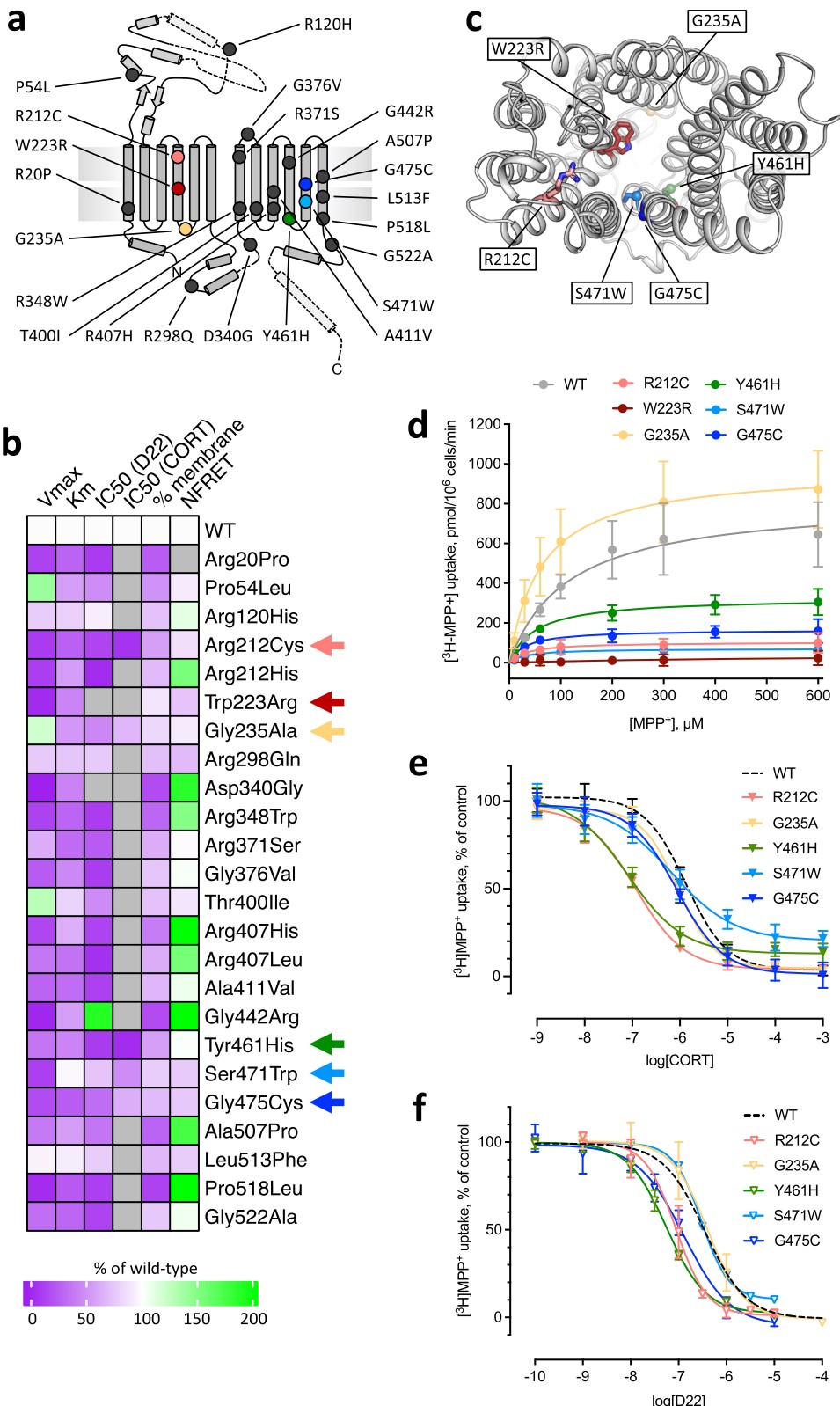

**Fig. 4 | Functional analysis of OCT3 mutants. a** Distribution of the OCT3 genetic variants, mapped on the structure of OCT3. **b** A heat map illustrating the effects of mutations. The scale bar indicates the increase (green) or decrease (magenta) of $V_{max}$, $K_m$, $IC_{50}$, membrane expression and NFRET compared to wild-type, as detailed in Materials and Methods. Arrows indicate the variants selected for detailed characterization. **c** View of OCT3 from the extracellular space, with the side-chains of the selected residues shown as sticks. **d** Uptake of MPP+ by the wild-type OCT3 (WT) and by the selected variants expressed in HEK293 cells (Supplementary Table 5). **e, f** Inhibition of MPP+ transport by CORT (**e**) and D22 (**f**). The WT is indicated with a dotted line. The values in **d**–**f** correspond to mean ± SD, $n = 3$–4 represents biologically independent experiments performed with three technical repeats.

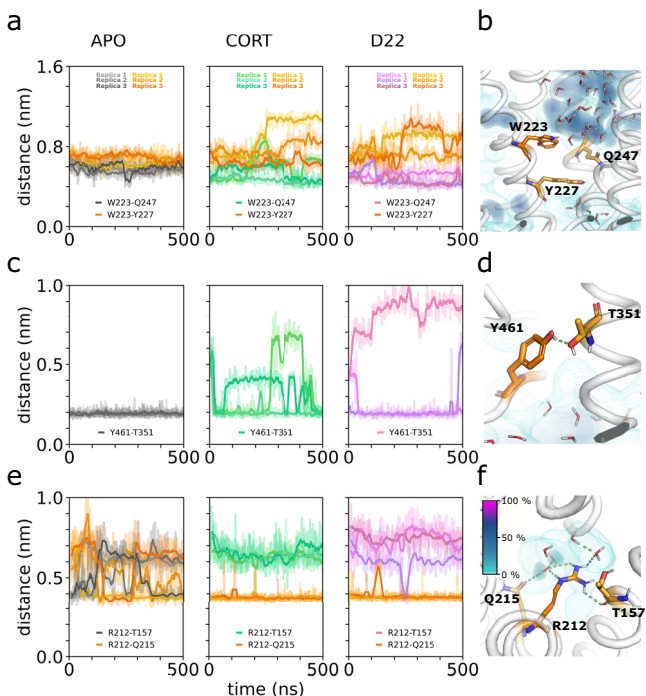

**Fig. 5 | The possible underlying causes of function disruption in genetic variants of OCT3. a** Distance between the side chains of W223 and Y227 or Q247, indicating the orientational dynamics of the sidechain of W223. **b** Local environment of W223, showing average water molecules occupancy in its proximity. **c** Donor-acceptor hydrogen bond distance between Y461 and T351, confirming the long-range effect of the bound inhibitor on the local structural stability of the Y461 site. **d** Zoom onto Y461, showing the hydrogen bond to T351 and averaged water occupancy in its proximity. **e** Distance of the side chain of R212 to residue T157 and Q215, highlighting the oscillation of the interaction pattern in the apo state compared to the stable conformation in the inhibitor bound states. **f** Close-up of R212, emphasizing on the hydrogen bond network formed with residues Q215, T157 and water molecules. Averaged spatial occupancy of water is represented as a density colored according to the legend.

lipid bilayer with very low efficiency. Approaches employing tracer flux experiments including cis-inhibition assays will help address and hopefully solve the question of the mode of action of these compounds.

Several questions remain, in particular concerning the substrate translocation pathway of OCT3 and the role of different domains in its activity. OCT3 shows an obvious access path to the substrate binding site that is wide open to the extracellular side. However, some of the substrates of OCT3 are hydrophobic, which may suggest accumulation in the lipid bilayer. It is tempting to speculate that OCT3 might be able to retrieve its substrates from the membrane, which is a mechanism well described for ABC transporters[39]. The OCT3 structure provides a hint for the location of such a lateral access site (the V-site), which remains to be validated. The role of the ECD of OCT3 is currently unexplored, and the mutations located in the ECD did not substantially affect transport properties of OCT3. However, by analogy with other MFS transporters that contain an extended ECD[40,41] this domain in OCTs may play an important role in molecular gating or in protein-protein interactions. For example, ECD could play a role in recruitment of CD63, a binding protein known to interact with OCTs, and to regulate their trafficking to the plasma membrane or intracellular compartments[42]. Moreover, it has been shown that both OCT1 and OCT2 depend on an intact ECD for oligomerization and trafficking to the plasma membrane[43,44].

The extended list of novel genetic variants functionally characterized allows us to place missense mutations with clear functional

consequence in the context of an experimentally determined OCT3 structure. Genetic variants located in certain regions of the protein clearly impact transport and/or surface expression of OCT3 and thus impact uptake of physiological substrates and drugs. Moreover, as therapeutic strategies involving OCTs emerge in a number of different areas, a better understanding of how genetic variations affect transporter function may be useful to understand personalized variations in drug response and to develop effective pharmacogenomics-based therapeutic strategies. Thus, the deeper knowledge of the structure and function of OCTs brings us closer to more precise and efficient medical treatments[22,35,36]. Therefore, our structures provide a starting point for rational drug development targeting OCT3 in the treatment of depression[22,45], diabetes[11,12,46], cardiac disease[10,15], and cancer chemotherapy[19,47].

## Methods
### Materials and chemicals
Detergents, dodecylmaltoside (DDM) and glyco-diosgenin (GDN), and Brain Polar Lipids were purchased from Antrace Inc. Decynium-22 was ordered from Synthon Chemicals (Bitterfeld-Wolfen, Deutschland). All other chemicals and cell culture supplies were obtained from Sigma-Aldrich (St. Louis, MO, USA) and Sarstedt (Nuembrecht, Germany), unless indicated otherwise.

### Cell lines and cell culture
hOCT3 wildtype plasmid was generously provided by Eric Gouaux, Vollum institute, Oregon Health & Science University, Portland, OR, USA. Fluorescently tagged constructs (YFP and CFP) were generated by cloning hOCT3cDNA to peCFP-C1 and peYFP-C1 (clontech) respectively. QuikChange site-directed mutagenesis kits and Quik-Change Primers (Agilent Technologies, Santa Clara, USA) were used to create plasmids of hOCT3 single nucleotide polymorphisms. Constructs were verified by sequencing (LGC Genomics, Berlin, Germany). Stable polyclonal HEK293 cell lines, expressing hOCT3 wildtype or hOCT3 genetic variants, were established as follows: HEK293 cells were transfected with the respective plasmids with the jetPRIME transfection method, according to the manufacturer's instructions (VWR International GmbH, Vienna, Austria). High selection pressure was maintained for 10 days by adding 100 µL geneticin (G418, 50 mg x mL$^{-1}$). 500,000 cells were then FACS sorted and polyclonal cell lines established, according to expression levels.

In cell culture, cells were maintained in high glucose- (4.5 g x L$^{-1}$), l-glutamine-containing (584 mg x L$^{-1}$) Dulbecco's Modified Eagle Medium (Sigma-Aldrich, St. Louis, USA) with 10% heat-inactivated Fetal Bovine Serum (FBS, Sigma-Aldrich), penicillin (1 U x mL$^{-1}$, Sigma-Aldrich) and streptomycin (1 µg x mL$^{-1}$, Sigma-Aldrich) added. For maintaining the selection process, geneticin (50 µg x mL$^{-1}$) was added regularly. Cells were maintained in 10-cm cell culture dishes (Greiner) at 37 °C and 5% CO$_2$ in an incubator. The day before uptake and uptake inhibition assays, cells were seeded onto PDL (poly-D-lysine) coated 96-well plates at a density of ~0.4 × 10$^5$ cells per well. For live confocal microscopy, cells were seeded onto PDL-coated 35 mm glass-bottom dishes (Cellvis, Sunnyvale, California, USA) at a density of 0.3 × 10$^6$ cells per dish.

### Radiotracer uptake assays
On the day of the experiment, cells were incubated with 50 µL Krebs-HEPES-buffer (KHB; 10 mM HEPES, 120 mM NaCl, 3 mM KCl, 2 mM CaCl$_2$, 2 mM MgSO$_4$ and 20 mM D-glucose, pH adjusted to 7.3) containing increasing concentrations of 1-methyl-4-phenylpyridinium (MPP$^+$; Sigma-Aldrich, St. Louis, MO, USA) together with 50 nM [$^3$H]-MPP$^+$ (80–85 µCi mmol$^{-1}$; American Radiolabeled Chemicals, St. Louis, USA) for 10 min. Time-dependent uptake was determined using 50 nM [$^3$H]-MPP$^+$ for the time indicated. Unspecific uptake was determined in the presence of 100 µM decynium-22. Cells were washed with KHB and

then lysed with 100 μL 1% sodium dodecyl sulfate (SDS). Lysate was transferred to a counting vial, containing 2 mL of scintillation cocktail. Uptake of tritiated substrate was assessed with a beta scintillation counter (Perkin Elmer, Waltham, USA).

## Radiotracer uptake inhibition assays

Cells were preincubated with vehicle or increasing concentrations of decynium-22 or corticosterone (dissolved in DMSO), diluted in 50 μL of KHB for 10 min. Preincubation solution was aspirated and uptake solution, containing KHB, vehicle or substance of interest at the desired concentration and 50 nM [$^3$H]-MPP$^+$, was added to the wells. Uptake was terminated by aspiration and cell washing with 100 μL ice-cold KHB after 10 min. Cells were subsequently lysed with 100 μL of 1% SDS. Lysate was transferred to a counting vial, containing 2 mL of scintillation cocktail. Uptake of tritiated substrate was assessed with a beta scintillation counter (Perkin Elmer, Waltham, USA).

## Confocal microscopy and image analysis

Confocal microscopy images were taken on a Nikon A1R + laser scanning confocal microscope system with a 60× NA 1.4 oil immersion objective (Nikon, Vienna, Austria). Cell culture medium was aspirated and cells incubated with trypan blue (0.4%, Sigma Aldrich) for 10 min and then washed with KHB multiple times. Cells were kept on KHB throughout the experiment.

eYFP Fluorescence was excited with a 488 nm, trypan blue with a 561 nm laser line. Emitted light was filtered, using a 525/50 nm (eYFP) or 595/50 nm (trypan blue) emission filter respectively and detected by a high-sensitivity GaAsP PMT detector. Three images were taken on three separate days. Image analysis was conducted in Fiji ImageJ 1.53c[1]. For analysis of membrane expression, in each image two regions of interest were drawn per cell, one encompassing the cell membrane (as defined by trypan blue staining) and one the cell interior. Membrane expression levels were defined as the relative values of membrane versus intracellular mean intensity.

## Immunoblotting

Cell lysates from heterologous HEK293 cells stably expressing YFP-tagged OCT3 proteins were solubilized in lysis buffer containing 1% Triton X-100, 20 mM Tris-HCl pH 8.0, 150 mM NaCl, 1 mM EDTA, 1 mM sodium orthovanadate, 5 mM sodium fluoride, 5 mM sodium pyrophosphate, and a protease inhibitor mixture (Roche Applied Science) on a tube rotator for 2 h at 4 °C. After centrifugation at 14,000 x g for 30 min at 4 °C, the supernatant was collected for immunoblotting. Proteins were separated on SDS-PAGE gels and transferred onto nitrocellulose membranes (GE healthcare), which were then blocked and immunostained with a rabbit anti-GFP polyclonal antibody (A6455, Thermo Fisher).

## Förster Resonance Energy Transfer imaging

Förster Resonance Energy Transfer (FRET) microscopy was utilized to assess protein-protein-interaction of OCT3 isoforms. Experiments were conducted using HEK293 cells transiently expressing OCT3-constructs labeled with a fluorescence donor (eCFP) and acceptor (eYFP) attached to the N-terminus, respectively. The cells were seeded into 29 mm dishes with 20 mm bottom (# 1.5 glass; Cellvis, Mountain View, CA, USA) at a density of 10$^5$ cells per dish 1 day prior to imaging. FRET was measured with an iMIC inverted microscope (T.I.L.L. Photonics GmbH, Kaufbeuren, Germany) equipped with a 60X (N.A. 1.49) oil objective (Olympus).

Fluorescence was excited with a 100 W Xenon Lamp (Polychrome, T.I.L.L Photonics GmbH, Kaufbeuren, Germany). The excitation light was filtered through 436/20 nm (eCFP) or 514/10 nm (eYFP) excitation filters (Semrock, Rochester, NY, USA) and directed to the sample by a 442/514 dual line dichroic mirror (Semrock, Rochester, NY, USA). The emitted fluorescence light was filtered through a 480/40 nm and 570/

80 nm dual emission filter (Semrock, Rochester, NY, USA) to a beamsplitter unit (Dichrotom, T.I.L.L Photonics, Kaufbeuren, Germany). Emission light was separated according to fluorescence wavelength using a 515 nm dichroic mirror and channels (<515 nm & >515 nm) projected side by side onto an EMCCD chip (iXon Ultra 897 Andor, Andor Technology, Belfast, UK). Live Acquisition software (version 2.5.0.21; T.I.L.L Photonics GmbH, Kaufbeuren, Germany) was used for recording. Two images were taken per set (donor and acceptor emission after donor excitation and acceptor emission after acceptor excitation). Per condition, ten sets were recorded on each of three experimental days and images were then analyzed using Offline Analysis software (version 2.5.0.2; T.I.L.L Photonics GmbH, Kaufbeuren, Germany). Background fluorescence was subtracted from each image, and one region of interest (part of the plasma membrane) per cell was selected in the CFP channel. The average intensity of each region of interest was used for calculations. HEK293 cells expressing a CFP or YFP signal only were used to determine spectral bleed through (BT) for donor (0.57) and acceptor (0.04). Normalized FRET (NFRET) was calculated as follows

$$NFRET = \frac{I_{FRET} - BT_{Donor} \times I_{Donor} - BT_{Acceptor} \times I_{Acceptor}}{\sqrt{I_{Donor} \times I_{Acceptor}}} \quad (1)$$

## Genetic data

Exome sequencing data were provided by the integrated psychiatric research (iPSYCH) consortium[1]. The exome sequencing data used in this study includes 19851 samples from the iPSYCH consortium's first phase genotyping of a nation-wide Danish birth cohort which has been described in greater detail previously[2]. The iPSYCH study sample is approved by the Danish Data Protection Agency. Informed consent is not required by law for register-based research in Denmark. Procedures for exome sequencing, sample and variant quality control was performed as described in ref. 3.

## Data and statistical analysis

IC$_{50}$, V$_{max}$ and K$_m$ values were calculated and graphs plotted with GraphPad Prism 9.2.0 (GraphPad Software Inc., San Diego, USA). Half maximal inhibitory concentrations (IC$_{50}$) were determined by non-linear regression, solving equation $Y = \text{Bottom} + (\text{Top-Bottom})/(1 + 10^{\wedge}(X\text{-LogEC50}))$. Michaelis-Menten kinetics were determined via $Y = V_{max}*X/(K_m + X)$. All data are from at least three biologically independent experiments ($n \geq 3$), in triplicate and portrayed as mean ± SD. Confocal and FRET microscopy images were taken and Western blot lysates prepared from three individual passages on three separate days. The heatmap was created in RStudio (2021.09.2), package: ComplexHeatmap. Fisher's-exact test was used to compare carrier frequencies of coding SLC22A3 variants in cases and controls.

## Protein expression and purification

The full-length human OCT3 (Uniprot: O75751) was cloned into pACMV-based vector with C-terminal 3C-YFP-TwinStrep fusion tag. The plasmids were transfected into HEK 293 F cells and a stable monoclonal cell line (HEK293F-hOCT3) capable of expressing OCT3 was generated for large scale expression. HEK293F-hOCT3 cells were grown in suspension at 37 °C to a density of ~3–3.5 × 10$^6$ mL$^{-1}$ in Gibco® FreeStyle™ 293 Expression Medium (ThermoFisher Scientific). The cells were collected by centrifugation at 800 g for 20 min and stored at −80 °C until the day of the experiment.

For purification, the cell pellets collected from to 0.5 L of cell culture were thawed and resuspended in buffer A (50 mM Tris-HCl, 150 mM NaCl, pH 8.0) supplemented with protease inhibitors (1 mM benzamidine, 1 μg/mL leupeptin, 1 μg/mL aprotinin, 1 μg/mL pepstatin, 1 μg/mL trypsin inhibitor and 1 mM PMSF) and lysed using a Dounce homogenizer. The lysate was centrifuged at 186,000 x $g$ (Ti45 rotor)

for 45 min, and membrane pellet was solubilized in the buffer B (50 mM Tris-HCl, 150 mM NaCl, 10% glycerol, 1% dodecylmaltoside (DDM) and 0.02% cholesteryl hemisuccinate (CHS), pH 8.0) for 1 h at 4 °C. The lysate was cleared by ultracentrifugation (Ti45 rotor, 186,000 x g for 30 min to remove insoluble debris). The supernatant was incubated with 4 mL of CNBr-activated Sepharose coupled to an anti-GFP nanobody[4]. After a 60 min incubation at 4 °C the resin was collected in a gravity column and washed with 40 column volumes of buffer C (50 mM Tris-HCl 150 mM NaCl, 0.02% glycol-diosgenin (GDN), 5% glycerol, pH 8.0). The protein was eluted using cleavage by HRV 3 C protease (1:10 w/w). The eluted protein was concentrated with a 30 kDa cut-off Amicon Ultra spin (Millipore) and subjected to size-exclusion chromatography (SEC) using a Superpose 6 Increase 10/300 GL column (GE Healthcare) equilibrated in buffer D (50 mM Tris-HCl, 150 mM NaCl, 0.02% GDN, pH 8.0). The fractions corresponding to purified hOCT3 (elution volume, 14.8–16.9 mL) were collected and pooled. The protein purity was assessed using 4–20% SDS-PAGE (BIO-RAD, Switzerland) and visualized by standard Coomassie brilliant blue staining technique.

**GFP-nanobody.** The expression and purification of GFP nanobody was carried out as previously described[4]. In short, the anti-GFP nanobody plasmid was transformed into *E. coli* BL21 (DE3) and grown in Luria Broth (LB) media supplemented with 50 µg/ml ampicillin at 37 °C. Once the optical density at 600 nm (OD600) of bacterial culture reached 0.5, the expression of protein was induced by adding 0.5 mM IPTG followed by overnight incubation at the 20 °C. Bacterial cultures were harvested at 4000 g for 20 min at 4 °C, and pellets were frozen in liquid N2 and stored at −80 °C. Frozen pellets were thawed on ice, re-suspended in buffer E (25 mM HEPES pH 8.0, 150 mM NaCl, 10 mM Imidazole, 1 mM PMSF, 10 ug/mL DNase I). The cells were lysed by sonication, and the cell lysate was centrifuged for 30 min at 20000xg. Clarified lysate was then incubated with Ni-NTA resin (1–2 mL bed volume of resin per 1 L of culture) for 30–40 min, and subsequently washed with 20 CV of buffer F (25 mM HEPES pH 8.0, 150 mM NaCl) supplemented with 50 mM imidazole, and later protein was eluted with 5 CV buffer F containing 250 mM Imidazole. The elution was concentrated with a 10 kDa cut-off Amicon concentrator and loaded on a Superdex 75 16/600 GL column (GE Healthcare) in buffer F. SEC fraction corresponding to GFP-nanobody was pooled and flash frozen in liquid N2, and stored at −80 °C. Protein concentration was determined by absorbance at 280 nm using $\varepsilon = 27055\ \mathrm{cm^{-1}M^{-1}}$.

**Membrane Scaffold Protein.** The expression and purification of Membrane Scaffold Protein MSP1D1 was carried out as previously described[5]. In brief, the MSP1D1 plasmid was transformed into *E. coli* BL21 (DE3) and grown at 37 °C in Terrific Broth (TB) media. The expression of protein was induced with 1 mM IPTG at OD600 of ~2–3 and incubated for 3 h. After harvesting by centrifugation, cell pellets was re-suspended in lysis buffer (50 mM Tris-HCl, 200 mM NaCl, 25 mM Imidazole, 1% Triton-X100, 1 mM PMSF and 10 ug/ml DNase I) and lysed by sonication. The clarified lysate after centrifugation (20,000 x *g* for 30 min) was incubated with Ni-NTA resin for 30 min, and subsequently step washed with 10 CV of buffer G (50 mM Tris-HCl pH 8.0, 150 mM NaCl) containing 25 mM Imidazole, 1% Triton-X100, 5 CV of buffer H (50 mM Tris-HCl pH 8.0, 150 mM NaCl, 25 mM Imidazole, 2% Sodium Cholate), 5 CV of buffer I (50 mM Tris-HCl pH 8.0, 150 mM NaCl, 50 mM Imidazole). Finally the protein was eluted in buffer I supplemented with 350 mM Imidazole. Elution fractions containing MSP1D1 were pooled, desalted in buffer J (20 mm Tris-HCl pH 8.0, 200 mM NaCl) and flash frozen in liquid N2, and stored at −80 °C. Protein concentration was determined by absorbance at 280 nm using $\varepsilon = 21430\ \mathrm{cm^{-1}M^{-1}}$.

### Reconstitution in lipid nanodisc
Reconstitution of OCT3 in MSP1D1 nanodiscs was performed using freshly purified transporter. In brief, 2.5 mg of brain polar lipid (BPL, Avanti) dissolved in 100 µL of chloroform was dried under a stream of N$_2$. Dried film of lipid was mixed with 300 µL of 3% DDM and the lipid-detergent mixture was then sonicated using bath sonicator (Bandelin SONOREX™ SUPER, Germany) until the mixture turned translucent. Detergent-solubilized BPL extract was added to freshly purified OCT3 at a molar ratio of 1: 75, and incubated for 30 min at room temperature with rotation. MSP1D1 was added to protein-lipid mixture and incubated for additional 30 min. Concentrations of OCT3 for nanodisc reconstitution were in the 10–15 µM range, with a molar ratio of OCT3 to MSP1D1 to lipid of 1:1:75. Following the incubation period, nanodisc formation was triggered by adding 300 mg of wet Bio-beads (washed with 100% methanol and with Milli-Q water). This final reconstitution mixture was incubated at 4 °C for 16 h (overnight) with gentle mixing. The supernatant was cleared of the beads by letting the beads settle and removing liquid carefully with a pipette. Sample was spun for 10 min at 25000 x g using a bench-top Eppendorf centrifuge before loading onto a Superose 6 Increase 10/300 GL column equilibrated in 20 mM Tris-HCl, pH 8.0, 150 mM NaCl. The peak fractions corresponding to OCT3 in MSP1D1 (elution volume, 15.4-16.9 mL) were collected, concentrated with a 30 kDa cutoff Amicon concentrator and used for cryo-EM grid preparation.

### Cryo-EM sample preparation and data collection
The cryo-EM samples were prepared using freshly reconstituted OCT3-BPL-MSP1D1 nanodisc samples. The complexes of OCT3-D22 and OCT3-Corticosterone were prepared by adding D22 (100 mM stock concentration in 100% DMSO) or CORT (50 mM stock concentration in 100% DMSO) at a final concentration of 1 mM, and incubating the samples on ice for 10–15 min. A final concentration of OCT3 and OCT3-drug complexes used for freezing grids was ~6-7 mg/mL, estimated based on absorbance at 280 nm and normalized extinction coefficient of 133700 M$^{-1}$ cm$^{-1}$ (considering two molecules of MSP1D1 per one OCT3 molecule).

Quantifoil 1.2/1.3 grids (300 mesh) were briefly glow-discharged in a PELCO easiGlow (Ted Pella) glow discharge cleaning system, for 25 s at 30 mAmp in air. A small amount of protein or protein-drug sample (3.5 µL) was then applied to glow-discharged grids, blotted for 3 s with blot force 20, and plunged into liquid ethane using Vitrobot Mark IV (Thermo Fisher Scientific) with 100% humidity at 4 °C. The frozen grids were transferred under cryogenic conditions and stored in liquid nitrogen for subsequent cryo-EM data collection.

### Cryo-EM data collection and processing
A total of four datasets with 5580, 12251, 6191, 5227 movies were collected using EPU on a 300 kV Titan Krios (Thermo Fisher Scientific) equipped with a Gatan K3 direct electron detector and a Gatan Quantum-LS GIF at ScopeM, ETH Zurich. All movies were acquired in super-resolution mode with a defocus range of −0.5 to −3 µm and a final calibrated pixel size of 0.33 Å. The total dose per movie was 48, 48, 48 and 49 e-/Å$^2$ for datasets 1, 2, 3, and 4, respectively. The cryo-EM processing was performed in Relion 3.1.2[6]. The flow chart of cryo-EM processing of apo-OCT3 is depicted in Figure S1. In brief, all movie stacks were motion-corrected using MotionCorr 1.1.0[7] and binned two-fold. All micrographs were CTF corrected using Gctf[8]. A total of 513 particles were manually picked and 2D classified to generate autopicking templates. Selected 2D classes from manually picked particles were used to autopick particles from all micrographs in dataset 1. After multiple rounds of 2D clarifications, the best set of 2D classes were used to generate an initial model to be used in 3D classification. The 3D projections from the best 3D class from dataset 1 were used as templates for autopicking of particles in the datasets 2, 3 and 4. After multiple rounds of 2D and 3D clarifications, a set of 316323 particles

was selected for masked 3D refinement (using masks including or excluding the nanodisc density), resulting in a 3D reconstruction at 3.65 Å resolution. The refined particles were subjected to another round of 3D classification without alignment and with masking of the nanodisc. The particles from the best 3D class were subjected to several iterative cycles of 3D refinement, CTF refinement and particle polishing, yielding a final post-processed density map at 3.2 Å resolution.

The data collection strategy and image processing of OCT3-drug complexes was similar to that of OCT3 apo. For the OCT3-D22 complex, three datasets of 11340, 15456, 29004 movies were collected and processed. The movies for dataset 1 were recorded with a total dose of 51 e-/Å², dataset 2 and 3 with a dose of 49 e-/Å². For OCT3-CORT complex, a total of three datasets with 23504, 10988, 15035 movies were collected, with total dose of 49, 55, 61 e-/Å², respectively. The movie stacks of OCT3-D22 dataset 3 and all datasets of OCT3-CORT were binned 2-fold during data acquisition in EPU. The detailed steps of OCT3-D22 and OCT3-CORT processing are shown in Supplementary Fig. 3 and 4, respectively.

### Model building and refinement

Model building of OCT3 was performed in Coot[9] using the final post-processed density maps. All maps showed good quality in the membrane-embedded portion of the transporter which, along with a SwissModel[10] generated homology model of the OCT3 TM region (using PDB ID: 6h7d as a template), greatly facilitated model building based on the protein sequence. The ectodomain of OCT3 was partially resolved and thus could not be completely modeled experimentally. A complete hybrid model of OCT3, including the ectodomain and all regions missing in the density map, was built using a model generated by AlphaFold (AF-O75451-F1)[11], joining the missing loops with the model built into the density map. The pLDDT scores for regions modeled using AlphaFold are tabulated in Supplementary Table 2. The ligands, D22 and CORT, were generated from SMILE codes using eLBOW[12]. The structures were refined using phenix.real_space_refine[13]. The quality of the final models were assessed using MolProbity[14]. All figures were generated using PyMOL 2.5.2[15] and ChimeraX[16].

The homology models of OCT1, OCT2 and OAT1 were generated using SwissModel[10] with OCT3 as template. The sequence identity and modeling scores are tabulated in Supplementary Table 3.

### Molecular dynamics simulations

**System setup and preparation of the environment.** The experimentally solved cryo-EM structure of apo OCT3, as well as the hybrid models that include the AlphaFold generated loops of apo, D22-bound and CORT-bound OCT3 were embedded in a cholesterol-phospholipid containing bilayer. In addition, because of the uncertainty of placing CORT into the cryo-EM density, also models for the second possible orientation of CORT-bound OCT3 structures (CORT-flipped) were created. All glutamate and aspartate sidechains were protonated using their default protonation state, histidine residues were neutralized and protonated at their epsilon nitrogen. To ensure complete lipid mixing and an equilibration of the lipid environment surrounding OCT3, the systems were first simulated using the coarse grained (CG) representation of the MARTINI force field[17]. The simulation box (10.0 × 10.0 × 14.6 nm) harbors water, 150 mM NaCl, 70:30 mol% POPC:cholesterol lipid mixture. OCT3 was embedded using the insane procedure[18] and simulated for 4 μs, while applying position restraints on OCT3. The equilibrated environment was backmapped to an all atom representation[19], while the CG OCT3 was replaced with the original all-atom structure to remove spurious structural distortions introduced by the conversion procedures. The assembled system was relaxed using the membed[20] procedure to relax possible local atom overlaps between OCT3 and its environment. The chain ends at the gaps of the experimentally determined OCT3 structure were neutralized using capping groups (acetylation and methylation) to avoid termini-charge dependent effects.

**Molecular dynamics simulation.** The amber99sb-ildn force field was used to describe the proteins, water and ions[21]. Force field parameters of corticosterone as well as D22 were implemented by applying the general amber force field (GAFF)[22] and ACPYPE[23], partial charges have been calculated by the R.E.D. Server[24]. The membrane was described using slipid[25]. All atom MD simulations were performed using Gromacs 2021.3[26]. All systems were energy minimized and equilibrated in four steps that consist of 2.5 ns long simulations, while slowly releasing the position restrain forces acting on the Cα atoms as well as the ligand, if present (1000, 100, 10, 1 kJ/mol/nm). Initial random velocities were assigned independently to every system. Production simulations were performed for 1.0 μs for the apo transporter and 0.5 μs for the ligand bound OCT3, while simulations of CORT-flipped OCT3 were 150 ns long. Temperature was maintained at 310 K using the v-rescale ($\tau$ = 0.5 ps) thermostat[27] by separately coupling solvent, membrane, and protein plus ligand, if present. Semi-isotropic pressure coupling was applied using the Parrinello-Rahman barostat[28], using 1 bar and applying a coupling constant of 20.1 ps. Long range electrostatic interactions were described using the smooth particle mesh Ewald method[29] with a cutoff of 0.9 nm. The van der Waals interactions were described using the Lennard Jones potentials applying a cutoff of 0.9 nm. Long range correction for energy and pressure were applied. Coordinates of all atoms were recorded at every 25 ps. The Molecular Dynamics Parameters (MDP) files used during production can be found in the SI.

### Reporting summary

Further information on research design is available in the Nature Research Reporting Summary linked to this article.

## Data availability

The data that support this study are available from the corresponding authors upon reasonable request. The cryo-EM density maps have been deposited in the Electron Microscopy Data Bank (EMDB) under accession codes EMD-14716 (OCT3 apo), EMD-14725 (OCT3 corticosterone), and EMD-14728 (OCT3 decynium-22). The atomic coordinates have been deposited in the Protein Data Bank (PDB) under accession codes 7ZH0 (OCT3 apo), 7ZHA (OCT3 corticosterone), and 7ZH6 (OCT3 decynium-22). The raw data of time traces from MD simulation, the PDB coordinates (including the RMSF values) of all structures shown, the starting and final structures of all simulations as well as the MPD parameter file used for the Gromacs simulations are available at www.zenodo.org (https://doi.org/10.5281/zenodo.7182740). Source Data underlying Fig. 1b; 3e; and Supplementary Figs. 1d, e; 14; 15; 16b; 17b; and 18a, d are available as a Source Data file. Source data are provided with this paper.

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

## Acknowledgements

The research described in this publication was supported by the Vienna Science and Technology Fund (WWTF) [CS 15–033] (H.H.S.), Austrian Science Fund (FWF) doctoral program Molecular Drug Targets [W1232] (H.H.S.), stand-alone project P34670-B20 (H.H.S., T.S and V.M.K.) and doctoral program Neuroscience [DOC33-B27] (H.H.S. and J.M.), Theodor Körner Fonds 2020 (J.M.), Swiss National Science Foundation (SNSF; grant No. P400PM_191032, D.L.; Sinergia grant No. 198545, V.M.K.), and the Lundbeck Foundation (R303-2018-3540, FH). This project has received funding from the European Union's Horizon 2020 research and innovation program under the Marie Skłodowska-Curie grant agreement No 860954 (T.S.). We thank Michael Freissmuth and Richard Kammerer for critical comments on the manuscript. In addition, we thank the PSI EM Facility for their support: Emiliya Poghosyan and Elisabeth Müller-Gubler, as well as Miroslav Peterek and Bilal Qureshi (ScopeM, ETH Zurich) for their support in cryo-EM data collection. We also thank Pavel Afanasyev (CEMK, ETH Zurich) for his advice and help in cryo-EM data handling. In addition, we thank Spencer Bliven and Marc Caubet Serrabou (PSI) for their support in high performance computing. In addition, the results presented have, in part, been achieved using the Vienna Scientific Cluster.

## Author contributions

B.K., J.M., F.H., R.G., E.L., D.L., T.S., V.M.K., H.H.S. designed experiments, analysed data and prepared figures. B.K., J.M., F.H., R.G., E.L., D.L. J.-W.Y., C.Q., M.H., K.J., O.K., K.S. performed experiments and analysed data. T.W. contributed to conception and critical revisions. U.G., T.S., V.M.K. and H.H.S. supervised the project. B.K., J.M., T.S., V.M.K. and H.H.S. wrote the manuscript with input from all co-authors.

## Competing interests

The authors declare no competing interests.
