## [Peer Review File · Nature Communications]

Structural basis of organic cation transporter-3 inhibitionReviewers' Comments:

Reviewer #1:

Remarks to the Author:

The manuscript entitled "Structural basis of organic cation transporter-3 inhibition" provides the first structure of an organic cation transporter. The importance of this work is indisputable; nonetheless, I have some concerns on the functional portion of the manuscript that must be addressed. To follow, some comments that might improve the overall quality of the data and of the discussion thereof.

⌋ The radiotracer uptake assay in HEK293 cells overexpressing OCT3 was performed with [3H]MPP+ at the extracellular concentrations of 50 nM with an incubation time of 10 minutes. I think it is important to show that MPP+ uptake is linear in this time range. Perhaps the authors could show the time course as part of figure S1.

⌋ It is unclear the design of the radiotracer uptake inhibition assay in HEK293 cells. The authors decided to pre-incubate the cells with the inhibitor (decynium-22 or corticosterone), followed by 10-min co-incubation with [3H]MPP+ at the extracellular concentrations of 50 nM. In my eyes, this design complicates the interpretation of the data, as the inhibitor/substrate is present at both side of the plasma membrane with both cis-inhibition/trans-stimulation contributing to the calculated IC50. This promiscuous effect might provide an explanation for the stimulatory effect of cortisone on MPP+ OCT1-mediated transport.

⌋ Along this line, in the discussion, the authors stated "it remains to be determined whether OCTs are able to move their inhibitors across the membrane" and that "Our structures show that OCT inhibitors may sterically block the translocation pathway by wedging themselves into a binding pocket deep within the 304 translocation pathway". The authors should consider performing heteroexchange assays where the efflux of [3H]MPP+ is measured in the presence or absence of inhibitor on the extracellular side. Should the inhibitor be transported across the plasma membrane, thus be a substrate, the efflux of [3H]MPP+ will be stimulated. This occurs because the outward-to-inward translocation rate of the ligand-OCT3 complex is slower in a zero-trans condition (no substrate at the opposite side of the plasma membrane) than in the infinite-trans condition (another substrate in excess at the opposite side of the membrane) when the cycling rate is maximized by the presence of substrates at both sides of the plasma membrane. Conversely, the inhibitor will inhibit the efflux of [3H]MPP+ by locking the carrier in the outward conformation or by depleting the unloaded carrier fraction accessible for [3H]MPP+ export as a result of a very low K_{off} . The latter, would represent the functional validation of the structure data.

⌋ In the result section, the authors presented the Molecular basis of OCT3 ligand specificity (page 5). I found confusing the statement "It is not surprising that OCT1, which has residues C36 (F36 in OCT3) and I446 (F450 in OCT3), binds D22 and CORT with lower affinity than OCT3, but with higher affinity compared to OCT2". I assume that the authors refer to figure 3e. However, my interpretation is that OCT1 and OCT2 have overlapping affinity for D22. No conclusion can be drawn on CORT, as OCT1 is stimulated by CORT, whereas OCT2 is inhibited. As mentioned in one of my previous point, proper cis-inhibition and trans-stimulation studies.

Reviewer #2:

Remarks to the Author:

OCT3 is an important pharmacological target and its dysregulation is implicated in human diseases and effects drug uptake. The authors present three cryoEM structures of the organic cation transporter OCT3 solved in the absence and presence of chemical inhibitors. The structures are consistent with the overall structural topology of other major facilitator transporters. An impressively extensive set of missense variants of the OCT3 gene were functionally characterized in cell-based assays to understand the structural basis for OCT3 function. MD simulations were used to aid in ligand docking when cryoEM densities were ambiguous and to probe how genetic variations affect channel function and inhibitor binding. Altogether, several residues were indicated to be important for substrate

translocation.

My concerns fall mainly on the quality of the cryoEM as presented and the approaches to modeling.

The displayed EM densities in Figure 1 do not appear to be consistent with the reported resolutions. At better than 3 Å in the TMDs, as reported in supplementary figures 2-4, the pitch of helices should be very apparent and almost all sidechain densities visible. It is very difficult to tell this quality in Figure 1. It would help to assess map resolvability if the lipids-nanodisc densities were removed and consider map sharpening. Given this difficulty in assessing the quality of the cryoEM map, I would also then suggest showing zoomed-in views of the representative densities for a variety of structural elements in the OCT3 maps, so that one can comprehend the model fit to density.

The authors note that several parts of the protein sequence were not able to be modeled by the cryoEM density alone and thus used AlphaFold to fill in the missing gaps. However, the quality of the map in these areas were not demonstrated and the sequence information is buried within the Figure S6 legend. An additional table indicating which OCT3 sequences were resolved by cryoEM and which were modeled by AlphaFold would be useful.

Additionally, a more detailed description of how AlphaFold derived portions of the model were assessed and incorporated into the final model is necessary. How well did AlphaFold predict the structure as determined by the pLDDT scores? How did the predicted model fit to the density, even at lower confidence values? Was the AlphaFold/hybrid structure refined to the density? Were the ligands also allowed to be refined in the ligand-bound models of OCT3?

I suggest adding to the previously requested table a column for min, max, average pLDDT scores and the predicted model's relative fit to density as indicated by a metric such as correlation to density or average map value for that region.

As a caveat, the final PDB model should be representative of the cryoEM data and while AlphaFold can fill in the missing structural elements, if the density is not apparent, I suggest that those regions should be represented by C-alpha/poly-A in the final PDB model.

The densities used to determine D22 fit to OCT3 density map are not presented. Provide additional details regarding how the ligand ascribed densities were identified from the entire map. Showing experimental EM densities are important to assess the fit and position of the ligands and to elute your point that the compounds are readily accommodated by the protein.

In regards to locating and improving the inhibitor densities have the authors considered looking into the occupancy of the ligand in the bound protein at the particle level, i.e. using a focused 3D classification approach to select for a subset of particles that have the ligand present? Additionally, this approach could be used to improve the resolvability of the Ecto-domain.

It would be good to show the lipids-ascribed densities in all of your OCT3 structures to support your statement (line 190) that the lipid binding sites are conserved

Minor points:

Line 109 - no figure 1j

Figure S6f- What is a "coronary dissection" with regards to a slice through a space-filled model of OCT3 in a membrane? Reconsider this choice of phrase.

Line 109-112: Figure 2a,b does not show that the compounds are readily accommodated by the protein - consider revising the sentence or figure.

Figure 2: I suggest labeling the helices in d,f

It's difficult to reconcile the orientations of the CORT in figure S7 with Figure S8.

While Figure S9 is impressive and consistent with the transporter's substrate preference, I am curious if the electrostatic potentials are displayed on the same scale since no scale bar indicating relative

charge was provided.

Provide PDB IDs for Fig3a-d

Please report confidence values pLDDTs for the OCT1 and 2 for Figure 3g. Moreover, there should be some clarification that the homology models are generated in the absence of ligand, which can affect side-chain orientations to accommodate ligands.

Line 236/240: typo 2D and 3D clarifications; should be classifications from which you eliminate certain classes.

Why use 6h7d as a model template? What is the sequence or structural homology?

Reviewer #3:

Remarks to the Author:

The manuscript by Khanppnavar et al describes the high-resolution cryo-EM structure of human OCT3 in the apo state and bound to two different inhibitors: corticosterone (CORT) and decynium-22 (D22). The structures show common features of MFS family. They incorporate modeling and molecular dynamics to build a more complete picture of the OCT3 structure. The authors also relate the functional characteristics of different human genetic variants to structural features and provide a basis for understanding the impact of OCT3 polymorphisms. I think the structures they provide are of interest to the researchers interested in MFS transporters in general. However, I do have some concerns about the manuscript that need to be addressed in terms of the soundness of the methodology as well as the presentation of results.

1. The authors claim the inhibitors like D22 and CORT not only bind to the substrate binding site and prevent the binding of the substrate, they also inhibit the outward-to-inward-facing conformational transition. What is the evidence for this claim?
2. A major issue with the data is that the molecular dynamics simulations do not show convergence. Fig. S6 shows that the RMSD keeps going up without reaching a plateau almost in all cases with or without the missing loops. This could indicate either a bad model or a short simulation. They need to extend the simulations until a qualitative plateau is reached or provide some justification on why the system keeps changing.
3. The authors state that the missing loops were generated using Alphafold. There is a brief explanation in the SI about the procedure. Did the Alphafold also have any effect on the parts that were not missing or it was solely used to generate missing regions? Also how close are the cry-EM structures to what AlphaFold2 would have predicted without using the cryo-EM structure as the template?
4. The authors state "CORT placement was assisted by molecular dynamics (MD) simulations". In the supporting information it is stated that "because of the uncertainty of placing CORT into the cryo-EM density, also models for the second possible orientation of CORT-bound OCT3 structures (CORT-flipped) were created". No further information is provided particularly when the differential binding orientation of D22 and CORT are discussed in the main paper (page 5), it is not clear which CORT model is considered.
5. Fig. 5 and its caption are confusing. (a) is supposed to be the χ_1 angle based on the caption but the y axis is distance. Also (c) is supposed to show "hydrogen bond" but it is not clear what they mean by that since again the y axis is a distance. Is it the donor-acceptor distance perhaps?
6. All resolved structures are in the outward-facing state. What is the reason for this? Does this indicate that the outward-facing state is the resting state of the protein or it has to do with the specific conditions of the experiment that favors the outward-facing state?

Point-by-point response:

Reviewer #1 (Remarks to the Author):

The manuscript entitled “Structural basis of organic cation transporter-3 inhibition” provides the first structure of an organic cation transporter. The importance of this work is indisputable; nonetheless, I have some concerns on the functional portion of the manuscript that must be addressed. To follow, some comments that might improve the overall quality of the data and of the discussion thereof.

We thank the Reviewer for their positive feedback and the constructive criticism we addressed below.

The radiotracer uptake assay in HEK293 cells overexpressing OCT3 was performed with $[^3\text{H}]\text{MPP}^+$ at the extracellular concentrations of 50 nM with an incubation time of 10 minutes. I think it is important to show that MPP^+ uptake is linear in this time range. Perhaps the authors could show the time course as part of figure S1.

RESPONSE: We agree with the reviewer and have added the time course experiments we conducted to Supplementary figure 1. It is evident that uptake is still in the linear range up to 10 minutes (see figure below, left panel; 10 min are indicated by the blue dotted line), afterwards a plateau is reached. This conclusion was also based by the fitting we provide in the figure (in green; only shown in this point-by-point response).

Due to the fact that 5 minutes of uptake time would have constituted a veritable alternative, we conducted an additional comparative uptake inhibition assay with 5- and 10-minutes uptake time. We saw very little difference (see the right panel of the figure below), and due to reasons of practicality – and the large amount of mutants we screened in a variety of assays (see heatmap, Fig. 4, Panel B), we chose 10 minutes as uptake time and kept these conditions consistent for all experiments. We are thus convinced that this allows for meaningful comparisons between variants and wild type OCT3.

Yet, we are well aware that large differences between substrates have been reported for the OCTs. Therefore, we intend to, in the future, compare various physiological substrate affinities and their effects on derived IC_{50} values in uptake inhibition assays between variants ^{1,2,3,4}.

} It is unclear the design of the radiotracer uptake inhibition assay in HEK293 cells. The authors decided to pre-incubate the cells with the inhibitor (decynium-22 or corticosterone), followed by 10-min co-incubation with [3H]MPP+ at the extracellular concentrations of 50 nM. In my eyes, this design complicates the interpretation of the data, as the inhibitor/substrate is present at both side of the plasma membrane with both cis-inhibition/trans-stimulation contributing to the calculated IC50. This promiscuous effect might provide an explanation for the stimulatory effect of cortisone on MPP+ OCT1-mediated transport.

RESPONSE: Concerning the interpretability of the uptake inhibition data, the current consensus in the field is that both corticosterone and D22 are non-transported inhibitors, thus we do not see any complications in data interpretation due to only substrate MPP+ being present intracellularly, and not the non-transported inhibitors.

We address the point of the Reviewer concerning corticosterone uptake inhibition at OCT1 below.

} Along this line, in the discussion, the authors stated "it remains to be determined whether OCTs are able to move their inhibitors across the membrane" and that "Our structures show that OCT inhibitors may sterically block the translocation pathway by wedging themselves into a binding pocket deep within the 304 translocation pathway". The authors should consider performing heteroexchange assays where the efflux of [3H]MPP+ is measured in the presence or absence of inhibitor on the extracellular side. Should the inhibitor be transported across the plasma membrane, thus be a substrate, the efflux of [3H]MPP+ will be stimulated. This occurs because the outward-to-inward translocation rate of the ligand-OCT3 complex is slower in a zero-trans condition (no substrate at the opposite side of the plasma membrane) than in the infinite-trans condition (another substrate in excess at the opposite side of the membrane) when the cycling rate is maximized by the presence of substrates at both sides of the plasma membrane. Conversely, the inhibitor will inhibit the efflux of [3H]MPP+ by locking the carrier in the outward conformation or by depleting the unloaded carrier fraction accessible for [3H]MPP+ export as a result of a very low K_{off}. The latter, would represent the functional validation of the structure data.

RESPONSE: We agree with the Reviewer that, at the current stage of scientific knowledge, the two quotes from our manuscript are speculative and might be misleading. We have rephrased the respective paragraph (page: 10, third paragraph of the discussion section) to:

"However, the current consensus in the field is that both corticosterone and D22 are non-transported inhibitors with scarce evidence that D22 may accumulate into astrocytes via an OCT3 dependent mechanism⁵. It remains to be determined whether OCTs are able to move inhibitors across the membrane. This property would be similar to multidrug transporters, such as ABCB1 and ABCG2, which are capable of transporting a variety of organic compounds, including their inhibitors elacridar and tariquidar. Our structures show that OCT3 inhibitors may sterically block the translocation pathway. Future studies will be necessary to determine whether these poses of the inhibitors completely impede transport, or whether the transporter is nevertheless capable of moving the inhibitor molecules across the lipid bilayer with very low efficiency."

Apart from rephrasing the current manuscript, we thank the author for the suggestion to embark on heteroexchange experiments. While we are interested in the method, we think that establishing it and conducting all the proposed experiments would broaden the scope of the already extensive manuscript too much. In addition, a comparison between mass spectrometry and heteroexchange determined status of substrate properties at OCT2 has recently been published: the authors concluded a "rather low sensitivity (26.7%) of the trans-stimulation assay for identifying OCT2 substrates, and caution with respect to the use of such assay may therefore be considered"⁶.

Indeed, we are – for obvious reasons – also intrigued by this question already for some time. Hence, for the last months, we have started to work on mass spectrometry-based intracellular quantification but have not yet reached the needed level of proficiency and therefore cannot include data in this manuscript. In addition, we combined these measurements with electrophysiological recordings either as two-voltage clamp in *Xenopus laevis* oocytes and patch-clamp recordings in the whole-cell mode (in HEK293 cells stably expressing OCT3). However, this study is definitely outside the scope of the present manuscript and we ask the reviewer for patience until we will be able to report on the results.

In the result section, the authors presented the Molecular basis of OCT3 ligand specificity (page 5). I found confusing the statement “It is not surprising that OCT1, which has residues C36 (F36 in OCT3) and I446 (F450 in OCT3), binds D22 and CORT with lower affinity than OCT3, but with higher affinity compared to OCT2”. I assume that the authors refer to figure 3e. However, my interpretation is that OCT1 and OCT2 have overlapping affinity for D22. No conclusion can be drawn on CORT, as OCT1 is stimulated by CORT, whereas OCT2 is inhibited. As mentioned in one of my previous point, proper cis-inhibition and trans-stimulation studies.

RESPONSE: It is, with critical distance, true, as the reviewer suggests, that differences in D22 affinities between OCT1 and OCT2 are overlapping and, thus, to be considered marginal. We have therefore decided to delete this sentence accordingly.

Concerning the corticosterone curve at OCT1, while the interpretation of the reviewer is one possibility, we have also frequently seen fluctuations like this (see figure below) in compounds previously shown not to interact (neither substrates nor inhibitors) with the OCTs at higher concentrations (and please bear in mind that only the three highest concentrations up to 300µM showed this behavior). In an attempt to clear this up, we have repeated the respective uptake inhibition assay four more times in triplicate and do not see the effect as pronounced anymore (see updated Figure 3, also inserted below).

Reviewer #2 (Remarks to the Author):

OCT3 is an important pharmacological target and its dysregulation is implicated in human diseases and effects

drug uptake. The authors present three cryoEM structures of the organic cation transporter OCT3 solved in the absence and presence of chemical inhibitors. The structures are consistent with the overall structural topology of other major facilitator transporters. An impressively extensive set of missense variants of the OCT3 gene were functionally characterized in cell-based assays to understand the structural basis for OCT3 function. MD simulations were used to aid in ligand docking when cryoEM densities were ambiguous and to probe how genetic variations affect channel function and inhibitor binding. Altogether, several residues were indicated to be important for substrate translocation.

We thank the Reviewer for their positive feedback and the constructive criticism we addressed below.

My concerns fall mainly on the quality of the cryoEM as presented and the approaches to modeling.

RESPONSE: The overall resolutions for each of the reconstructions are in fact consistent with the presented maps. While in the core of the maps we have higher resolution of the density, some regions of the maps are indeed poorly resolved. To ensure that we do not give the reader a wrong impression, we have now introduced new supplementary figures, highlighting the individual map elements and their quality (Supplementary Figures 6 and 8, see also below on the next pages).

Supplementary Fig. 6. Cryo-EM map features and model building of hOCT3. **a**, Isolated density map for 12 TM helices of apo-OCT3 contoured at 12 σ threshold level. **b**, Cryo-EM density features of ectodomain (ECD) contoured at 10 σ , 7 σ and 4 σ levels. **c**, Cryo-EM density features for intracellular regions/domains including N- and C-terminus of OCT3. The residues modelled using AlphaFold are colored black.

Supplementary Fig. 8. Modelling of D22 and CORT in density maps. a, Cartoon representation of OCT3 showing density of D22 (magenta mesh) at 10σ and 4σ contour levels. **b**, Cartoon representation of OCT3 showing density of CORT (green mesh) occupying the center of the substrate translocation pathway. The CORT molecule was modelled in two alternate conformations labelled as CORT and CORT-flipped (~180 flipped orientations of CORT).

The displayed EM densities in Figure 1 do not appear to be consistent with the reported resolutions. At better than 3 Å in the TMDs, as reported in supplementary figures 2-4, the pitch of helices should be very apparent and almost all sidechain densities visible. It is very difficult to tell this quality in Figure 1. It would help to assess map resolvability if the lipids-nanodisc densities were removed and consider map sharpening. Given this difficulty in assessing the quality of the cryoEM map, I would also then suggest showing zoomed-in views of the representative densities for a variety of structural elements in the OCT3 maps, so that one can comprehend the model fit to density.

RESPONSE: Figure 1 illustrates the overall 3D reconstructions, including the lipid densities. Although we could add multiple additional panels to this figure, we opted to reserve this for the supplementary figures where we have the space to elaborate on the quality of the 3D reconstructions. The new Supplementary figures 6 and 8 illustrate the quality of the map (see above). For the purpose of clear data presentation, we prefer to keep the Figure 1 as originally presented, but extend it with the Supplementary figures 6 and 8.

The authors note that several parts of the protein sequence were not able to be modeled by the cryoEM density alone and thus used AlphaFold to fill in the missing gaps. However, the quality of the map in these areas were not demonstrated and the sequence information is buried within the Figure S6 legend. An additional table indicating which OCT3 sequences were resolved by cryoEM and which were modeled by AlphaFold would be useful.

RESPONSE: Indeed, the poor quality of the ECD density precluded model building into the map. Therefore, we combined the model built using the experimental cryo-EM with the fragments of the model derived from the database AlphaFold entry (AF-O75751-F1). We have included an explicit description of the sequence ranges from the AlphaFold model used for completing the full “hybrid” model (Supplementary Table 2). This hybrid model was used in the MD simulations, as a likely approximation to the full model of OCT3. The pdb deposition was done using only the partial model that was derived from cryo-EM data (i.e., missing the unresolved protein sequence fragments).

Residues range	Number of residues	Modelling method	pLDDT score (AlphaFold)		
			Min	Max	Average
1-74	74	EXP	47.25	95.60	84.92
75-86	12	AF	39.82	79.19	52.57
87-99	13	EXP	56.85	86.22	77.70
100-113	14	AF	49.47	84.08	68.11
114-316	202	EXP	68.26	98.25	90.56
317-327	21	AF	53.93	86.17	67.45
328-535	207	EXP	37.85	97.71	85.91
536-556	21	AF	27.76	61.69	36.85

Additionally, a more detailed description of how AlphaFold derived portions of the model were assessed and incorporated into the final model is necessary.

RESPONSE: We used the database AlphaFold model, and we picked the regions of the model that could be integrated into our experimental model according to the low resolution density elements present in the cryo-EM map.

How well did AlphaFold predict the structure as determined by the pLDDT scores?

RESPONSE: AlphaFold predicted the structure reasonably well, based on the overall pLDDT scores of 83.8. The TM domains were predicted with higher pLDDT scores of 89.4 and reasonable pLDDT scores of 77.5 for ectodomain, and could be reasonably placed into the resolved part of ectodomain density map.

How did the predicted model fit to the density, even at lower confidence values?

RESPONSE: The predicted AlphaFold model of OCT3 was found to be in an inward open conformation compared to the experimentally-determined outward open conformation. As a result, fitting of the complete predicted model of OCT3 into the density map was not performed. However, the individual structural elements of the protein i.e. TM1-6 & TM7-12, and ECD could be separately placed into the density with higher accuracy.

Was the AlphaFold/hybrid structure refined to the density?

RESPONSE: We have not refined the hybrid model on experimental density map due to low resolution structural features in the unresolved regions.

Were the ligands also allowed to be refined in the ligand-bound models of OCT3?

RESPONSE: Yes, they were refined in our experimental models but not in the hybrid models.

I suggest adding to the previously requested table a column for min, max, average pLDDT scores and the predicted model's relative fit to density as indicated by a metric such as correlation to density or average map value for that region.

RESPONSE: We have included the pLDDT scores as Supplementary table 2. However, as we have not refined the hybrid model to the density due to very poor density quality, we are not reporting any metrics related to fit to density. This may be a meaningful action to perform in the future, contingent on resolving the ECD density better than what we have now.

As a caveat, the final PDB model should be representative of the cryoEM data and while AlphaFold can fill in the missing structural elements, if the density is not apparent, I suggest that those regions should be represented by C-alpha/poly-A in the final PDB model.

RESPONSE: We completely agree – the final deposited model only includes the parts of the protein that we could confidently build into the density. We took an even more conservative approach than suggested by the reviewer by excluding any additional residues in the unresolved regions of the map. The hybrid model is very useful for MD: it is stable in the performed simulations, which further validates the generated hybrid model, while the experimental only model misses part of the hydrophobic core of the ECD, which therefore is unstable in simulations.

The densities used to determine D22 fit to OCT3 density map are not presented. Provide additional details regarding how the ligand ascribed densities were identified from the entire map. Showing experimental EM densities are important to assess the fit and position of the ligands and to elute your point that the compounds are readily accommodated by the protein.

RESPONSE: We have generated the new Supplementary figure 8 illustrating density maps corresponding to D22 and CORT.

In regards to locating and improving the inhibitor densities have the authors considered looking into the occupancy of the ligand in the bound protein at the particle level, i.e. using a focused 3D classification approach to select for a subset of particles that have the ligand present?

Additionally, this approach could be used to improve the resolvability of the Ecto-domain.

RESPONSE: We have attempted focused classifications without alignment for ligands and ECD, but this did not produce any appreciable improvement in the density. We reasoned that the size of the map is quite small, therefore did not pursue this further.

It would be good to show the lipids-ascribed densities in all of your OCT3 structures to support your statement (line 190) that the lipid binding sites are conserved

RESPONSE: We have generated the new Supplementary figure 13, which illustrates the lipid positions.

Minor points:

Line 109 - no figure 1j

RESPONSE: This has been corrected.

Figure S6f- What is a “coronary dissection” with regards to a slice through a space-filled model of OCT3 in a membrane? Reconsider this choice of phrase.

RESPONSE: We thank reviewer for highlighting this issue. We have rephrased the sentence in revised manuscript and replaced “coronary dissection” with “slice through”.

Line 109-112: Figure 2a,b does not show that the compounds are readily accommodated by the protein - consider revising the sentence or figure.

RESPONSE: We have rephrased the sentence in the revised manuscript. We now changed the sentence to refer to Figure 2, as the whole figure (different panels) addresses the conformational changes and the ligand binding.

Figure 2: I suggest labeling the helices in d,f

RESPONSE: We included the labels of helices in Fig. 2d & f – in the right-most panel only, to avoid crowding of the labels. This labeling should allow the readers to orient themselves in the models without seeing too many labels all at once.

It's difficult to reconcile the orientations of the CORT in figure S7 with Figure S8.

RESPONSE: We thank reviewer for spotting this issue. We have clarified the confusion in the new figure legends of the revised manuscript, introducing “CORT” and “CORT-flipped” labels.

While Figure S9 is impressive and consistent with the transporter's substrate preference, I am curious if the electrostatic potentials are displayed on the same scale since no scale bar indicating relative charge was provided.

RESPONSE: We have included the appropriate description in the legend: “Electrostatic potential for each of the models was calculated with the Adaptive Poisson-Boltzmann Solver (APBS) module in Pymol and depicted on the same scale (-10 kT/e red, +10 kT/e blue).”

Provide PDB IDs for Fig3a-d

RESPONSE: We have provided the PDB IDs for the figure 3a-d in the figure legend.

Please report confidence values pLDDTs for the OCT1 and 2 for Figure 3g. Moreover, there should be some clarification that the homology models are generated in the absence of ligand, which can affect side-chain orientations to accommodate ligands.

RESPONSE: We have generated these models using the SwissModel server. We have included the relevant homology model scores in the Supplementary Table 3.

Name	Sequence identity (%)	Modelling score	
		GMQE	QMEAN DisCo Global
OCT1	51.52	0.47	0.50 ± 0.05
OCT2	51.87	0.47	0.48 ± 0.05
OAT1	35.31	0.44	0.47 ± 0.05

Line 236/240: typo 2D and 3D clarifications; should be classifications from which you eliminate certain classes.

RESPONSE: The typo has been fixed.

Why use 6h7d as a model template? What is the sequence or structural homology?

RESPONSE: 6h7d was one of top output structure obtained after blast of OCT3 protein sequence in RCSB database. The sequence similarity of OCT3 and SPT10 (PDB ID: 6h7d) was 31.3%, and visually the structure of 6h7d exhibited outward open conformation similar to OCT3. Therefore, it was used for the initial model building process.

Reviewer #3 (Remarks to the Author):

We thank the Reviewer for their positive feedback and the constructive criticism we addressed below.

The manuscript by Khanppnavar et al describes the high-resolution cryo-EM structure of human OCT3 in the apo state and bound to two different inhibitors: corticosterone (CORT) and decynium-22 (D22). The structures show common features of MFS family. They incorporate modeling and molecular dynamics to build a more complete picture of the OCT3 structure. The authors also relate the functional characteristics of different human genetic variants to structural features and provide a basis for understanding the impact of OCT3 polymorphisms. I think the structures they provide are of interest to the researchers interested in MFS transporters in general. However, I do have some concerns about the manuscript that need to be addressed in terms of the soundness of the methodology as well as the presentation of results.

1. The authors claim the inhibitors like D22 and CORT not only bind to the substrate binding site and prevent the binding of the substrate, they also inhibit the outward-to-inward-facing conformational transition. What is the evidence for this claim?

RESPONSE: The binding mode of D22 is incompatible with OCT3 reaching an inward-facing conformation, because its binding pose prevents the OCT3 domain to sufficiently associate at the extracellular side to allow for a conformational transition to the inward-facing conformation. For CORT, we observe additional densities of CORT molecules at the entrance of translocation shown in Supplementary figure 10. These CORT molecules may also hinder OCT3 reaching inward-facing conformation. Based on this structural evidence, we propose that the inhibitors may function by both blocking the substrate binding site and by preventing the outward-to-inward conformational transitions.

2. A major issue with the data is that the molecular dynamics simulations do not show convergence. Fig. S6 shows that the RMSD keeps going up without reaching a plateau almost in all cases with or without the missing loops. This could indicate either a bad model or a short simulation. They need to extend the simulations until a qualitative plateau is reached or provide some justification on why the system keeps changing.

RESPONSE: We extended the simulations of apo OCT3 from 0.5 to 1.0 μ s to verify that the RMSD values reached a plateau. The extended simulations confirmed that the hybrid model reached a stable plateau, while the incomplete OCT3 model was unstable, which consists of only the experimentally determined part of OCT3. A continuation of the drift of the incomplete OCT3 structural model should be expected, because the hydrophobic core of the ECD is incomplete and therefore structurally unstable.

3. The authors state that the missing loops were generated using AlphaFold. There is a brief explanation in the SI about the procedure. Did the AlphaFold also have any effect on the parts that were not missing or it was solely used to generate missing regions? Also how close are the cry-EM structures to what AlphaFold2 would have predicted without using the cryo-EM structure as the template?

RESPONSE: We have expanded the description of the AlphaFold model-assisted hybrid model building. The AlphaFold model did not influence the experimentally resolved regions of the model at all. The hybrid model was primarily used for MD simulations.

The predicted AlphaFold OCT3 model is in an inward-open state – this is true for OCT3, as well as for OCT1 and OCT2. As our experimentally-determined state is the outward-facing state, we have only a limited use for the AlphaFold model – although of course it is tempting to consider the predicted model as a truthful representation of

the inward-facing conformation of the transporter. This can be used in future MD simulations of the transporter going through the transport cycle.

4. The authors state "CORT placement was assisted by molecular dynamics (MD) simulations". In the supporting information it is stated that "because of the uncertainty of placing CORT into the cryo-EM density, also models for the second possible orientation of CORT-bound OCT3 structures (CORT-flipped) were created". No further information is provided particularly when the differential binding orientation of D22 and CORT are discussed in the main paper (page 5), it is not clear which CORT model is considered.

RESPONSE: We thank the reviewer for identifying this ambiguity. In the revised manuscript we have included the new Supplementary figure 8 (followed by a figure detailing our MD simulations in Supplementary figure 9) for clarifying the MD-guided modelling of particular CORT orientation in our final structure.

5. Fig. 5 and its caption are confusing. (a) is supposed to be the χ_1 angle based on the caption but the y axis is distance. Also (c) is supposed to show "hydrogen bond" but it is not clear what they mean by that since again the y axis is a distance. Is it the donor-acceptor distance perhaps?

RESPONSE We thank the reviewer for spotting this inconsistency. We have updated the incorrect figure caption. Panel (a) shows a distance, not an angle. Panel (c) shows the donor acceptor distance of the hydrogen bond.

6. All resolved structures are in the outward-facing state. What is the reason for this? Does this indicate that the

outward-facing state is the resting state of the protein or it has to do with the specific conditions of the experiment that favors the outward-facing state?

RESPONSE: This is a very important but still unresolved issue. The protein obviously adopts a stable outward facing conformation under these experimental conditions. We have yet to determine the conditions necessary to induce an occluded or an inward-facing state. It is possible that this depends on multiple factors, including buffer composition, pH, the presence of specific substrates or inhibitors that can preferentially stabilize the inward facing conformation. At this point we can only speculate why the observed conformation is outward facing. To avoid undue speculations, in the absence of a clear answer, we prefer to refrain from an in-depth discussion of this point in this manuscript.

References used in this document:

1. Sandoval PJ, Zorn KM, Clark AM, Ekins S, Wright SH. Assessment of Substrate-Dependent Ligand Interactions at the Organic Cation Transporter OCT2 Using Six Model Substrates. *Molecular Pharmacology* **94**, 1057 (2018).
2. Belzer M, Morales M, Jagadish B, Mash EA, Wright SH. Substrate-Dependent Ligand Inhibition of the Human Organic Cation Transporter OCT2. *Journal of Pharmacology and Experimental Therapeutics* **346**, 300 (2013).
3. Amphoux A, *et al.* Differential pharmacological in vitro properties of organic cation transporters and regional distribution in rat brain. *Neuropharmacology* **50**, 941-952 (2006).
4. Duan H, Wang J. Selective Transport of Monoamine Neurotransmitters by Human Plasma Membrane Monoamine Transporter and Organic Cation Transporter 3. *Journal of Pharmacology and Experimental Therapeutics* **335**, 743 (2010).
5. Inyushin M, *et al.* Membrane potential and pH-dependent accumulation of decynium-22 (1,1'-diethyl-2,2'-cyanine iodide) fluorescence through OCT transporters in astrocytes. *Bol Asoc Med P R* **102**, 5-12 (2010).
6. Lefèvre CR, *et al.* Substrate-Dependent Trans-Stimulation of Organic Cation Transporter 2 Activity. *International Journal of Molecular Sciences* **22**, 12926 (2021).

Reviewers' Comments:

Reviewer #1:

Remarks to the Author:
See attached file

Reviewer #2:

Remarks to the Author:
I am satisfied with the authors' response and I have no further concerns.

Reviewer #3:

Remarks to the Author:
The authors have addressed my concerns in a satisfactory manner.

Point-by-point response:

Reviewer #1 (Remarks to the Author):

RESPONSE R2: We thank the reviewer for their constructive criticism we addressed below.

The radiotracer uptake assay in HEK293 cells overexpressing OCT3 was performed with [3H]MPP+ at the extracellular concentrations of 50 nM with an incubation time of 10 minutes. I think it is important to show that MPP+ uptake is linear in this time range. Perhaps the authors could show the time course as part of figure S1.

RESPONSE: We agree with the reviewer and have added the time course experiments we conducted to Supplementary figure 1. It is evident that uptake is still in the linear range up to 10 minutes (see figure below, left panel; 10 min are indicated by the blue dotted line), afterwards a plateau is reached. This conclusion was also based by the fitting we provide in the figure (in green; only shown in this point-by-point response).

Due to the fact that 5 minutes of uptake time would have constituted a veritable alternative, we conducted an additional comparative uptake inhibition assay with 5- and 10-minutes uptake time. We saw very little difference (see the right panel of the figure below), and due to reasons of practicality – and the large amount of mutants we screened in a variety of assays (see heatmap, Fig. 4, Panel B), we chose 10 minutes as uptake time and kept these conditions consistent for all experiments. We are thus convinced that this allows for meaningful comparisons between variants and wild type OCT3.

Yet, we are well aware that large differences between substrates have been reported for the OCTs. Therefore, we intend to, in the future, compare various physiological substrate affinities and their effects on derived IC50 values in uptake inhibition assays between variants ^{1, 2, 3, 4}.

R#1: I thank the authors for the response and for providing supplementary data. However, considering that the data are removed of the WT values, the uptake is not linear either at 10 or at 5 min as the line does not intercept at the origin but rather high on the y-axis. In my experience, MPP+ uptake mediated by OCTs deflects from linearity within seconds.

RESPONSE R2: We fully agree that the uptake process mediated by OCTs is a quite difficult kinetic process. For the point we want to make, however, examining the uptake properties of the genetic OCT3-variants, we believe that our data are sufficiently valid to illustrate how the described variants (relatively) differ from the wildtype. We nevertheless take the reviewer's points as encouragement to proceed with the project outlined in the previous rebuttal letter.

It is unclear the design of the radiotracer uptake inhibition assay in HEK293 cells. The authors decided to pre-incubate the cells with the inhibitor (decynium-22 or corticosterone), followed by 10-min co-incubation with [3H]MPP+ at the extracellular concentrations of 50 nM. In my eyes, this design complicates the interpretation of the data, as the inhibitor/substrate is present at both side of the plasma membrane with both cis-inhibition/trans-stimulation contributing to the calculated IC50. This promiscuous effect might provide an explanation for the stimulatory effect of cortisone on MPP+ OCT1-mediated transport.

RESPONSE: Concerning the interpretability of the uptake inhibition data, the current consensus in the field is that both corticosterone and D22 are non-transported inhibitors, thus we do not see any complications in data interpretation due to only substrate MPP+ being present intracellularly, and not the non-transported inhibitors.

R#1: Well, based on the original discussion, it seems quite an open question for the authors as well, whether corticosterone and D22 are also substrates of OCT3. The new discussion has closed the issue. However, in consideration of the uncertainty in the issue, the authors should have taken the “conservative” experimental approach with a standard cis-inhibition assay, which can still be easily done.

RESPONSE R2: We respect the tenacious adherence – and suggestion of the reviewer – and we are intrigued to try this approach. However, we feel that this is not within the prime scope of the current manuscript – which is the structure and how genetic variants may support our understanding of the

structural and functional activity of OCT3. Therefore, we added the following sentence: “Approaches employing tracer flux experiments including cis-inhibition assays will help address and hopefully solve the question of the mode of action of these compounds.” (page 11, second to last paragraph).

We address the point of the Reviewer concerning corticosterone uptake inhibition at OCT1 below.

) Along this line, in the discussion, the authors stated “it remains to be determined whether OCTs are able to move their inhibitors across the membrane” and that “Our structures show that OCT inhibitors may sterically block the translocation pathway by wedging themselves into a binding pocket deep within the 304 translocation pathway”. The authors should consider performing heteroexchange assays where the efflux of [3H]MPP+ is measured in the presence or absence of inhibitor on the extracellular side. Should the inhibitor be transported across the plasma membrane, thus be a substrate, the efflux of [3H]MPP+ will be stimulated. This occurs because the outward-to-inward translocation rate of the ligand-OCT3 complex is slower in a zero-trans condition (no substrate at the opposite side of the plasma membrane) than in the infinite-trans condition (another substrate in excess at the opposite side of the membrane) when the cycling rate is maximized by the presence of substrates at both sides of the plasma membrane. Conversely, the inhibitor will inhibit the efflux of [3H]MPP+ by locking the carrier in the outward conformation or by depleting the unloaded carrier fraction accessible for [3H]MPP+ export as a result of a very low K_{off}. The latter, would represent the functional validation of the structure data.

RESPONSE: We agree with the Reviewer that, at the current stage of scientific knowledge, the two quotes from our manuscript are speculative and might be misleading. We have rephrased the respective paragraph (page: 10, third paragraph of the discussion section) to:

“However, the current consensus in the field is that both corticosterone and D22 are non-transported inhibitors with scarce evidence that D22 may accumulate into astrocytes via an OCT3 dependent mechanism⁵. It remains to be determined whether OCTs are able to move inhibitors across the membrane. This property would be similar to multidrug transporters, such as ABCB1 and ABCG2, which are capable of transporting a variety of organic compounds, including their inhibitors elacridar and tariquidar. Our structures show that OCT3 inhibitors may sterically block the translocation pathway. Future studies will be necessary to determine whether these poses of the inhibitors completely impede transport, or whether the transporter is nevertheless capable of moving the inhibitor molecules across the lipid bilayer with very low efficiency.”

Apart from rephrasing the current manuscript, we thank the author for the suggestion to embark on heteroexchange experiments. While we are interested in the method, we think that establishing it and conducting all the proposed experiments would broaden the scope of the already extensive manuscript too much. In addition, a comparison between mass spectrometry and heteroexchange determined status of substrate properties at OCT2 has recently been published: the authors concluded a “rather low sensitivity (26.7%) of the trans-stimulation assay for identifying OCT2 substrates, and caution with respect to the use of such assay may therefore be considered”⁶.

*Indeed, we are – for obvious reasons – also intrigued by this question already for some time. Hence, for the last months, we have started to work on mass spectrometry-based intracellular quantification but have not yet reached the needed level of proficiency and therefore cannot include data in this manuscript. In addition, we combined these measurements with electrophysiological recordings either as two-voltage clamp in *Xenopus laevis* oocytes and patch-clamp recordings in the whole-cell mode (in HEK293 cells stably expressing OCT3). However, this study is definitely outside the scope of the present manuscript and we ask the reviewer for patience until we will be able to report on the results.*

R#1: One comment, just for the sake of idea exchange, on the low sensitivity of the trans-stimulation. This might be due to the experimental design chosen by the authors of that paper. The authors pre-loaded the cells with the X candidate. Unfortunately, they cannot control the intracellular concentration of X, meaning that X can be in the cells but not in a concentration high enough to transstimulate the uptake of the tracer. The experiment should be performed the other way around. Loading with the tracer and measure its efflux in the presence of X on the outside, where X concentration is the independent variable.

RESPONSE R2: We thank the reviewer for their insightful suggestion – and will apply it in future experiments to quantify efflux, trans-stimulation and trans-inhibition.

) In the result section, the authors presented the Molecular basis of OCT3 ligand specificity (page 5). I found confusing the statement “It is not surprising that OCT1, which has residues C36 (F36 in OCT3) and I446 (F450 in OCT3), binds D22 and CORT with lower affinity than OCT3, but with higher affinity compared to OCT2”. I assume that the authors refer to figure 3e. However, my interpretation is that OCT1 and OCT2 have overlapping affinity for D22. No conclusion can be drawn on CORT, as OCT1 is stimulated by CORT, whereas OCT2 is inhibited. As mentioned in one of my previous point, proper cis-inhibition and trans-stimulation studies.

RESPONSE: It is, with critical distance, true, as the reviewer suggests, that differences in D22 affinities between OCT1 and OCT2 are overlapping and, thus, to be considered marginal. We have therefore decided to delete this sentence accordingly.

Concerning the corticosterone curve at OCT1, while the interpretation of the reviewer is one possibility, we have also frequently seen fluctuations like this (see figure below) in compounds previously shown not to interact (neither substrates nor inhibitors) with the OCTs at higher concentrations (and please bear in mind that only the three highest concentrations up to 300µM showed this behavior). In an attempt to clear this up, we have repeated the respective uptake inhibition assay four more times in triplicate and do not see the effect as pronounced anymore (see updated Figure 3, also inserted below).

R#1: Thank you for repeating the experiment.

RESPONSE R2: We thank the reviewer for their critical view – and were happy to have repeated it.

References used in this document:

1. Sandoval PJ, Zorn KM, Clark AM, Ekins S, Wright SH. Assessment of Substrate-Dependent Ligand Interactions at the Organic Cation Transporter OCT2 Using Six Model Substrates. *Molecular Pharmacology* **94**, 1057 (2018).
2. Belzer M, Morales M, Jagadish B, Mash EA, Wright SH. Substrate-Dependent Ligand Inhibition of the Human Organic Cation Transporter OCT2. *Journal of Pharmacology and Experimental Therapeutics* **346**, 300 (2013).
3. Amphoux A, et al. Differential pharmacological in vitro properties of organic cation transporters and regional distribution in rat brain. *Neuropharmacology* **50**, 941-952 (2006).
4. Duan H, Wang J. Selective Transport of Monoamine Neurotransmitters by Human Plasma Membrane Monoamine Transporter and Organic Cation Transporter 3. *Journal of Pharmacology and Experimental Therapeutics* **335**, 743 (2010).
5. Inyushin M, et al. Membrane potential and pH-dependent accumulation of decynium-22 (1,1'-diethyl-2,2'-cyanine iodide) fluorescence through OCT transporters in astrocytes. *Bol Asoc Med P R* **102**, 5-12 (2010).
6. Lefèvre CR, et al. Substrate-Dependent Trans-Stimulation of Organic Cation Transporter 2 Activity. *International Journal of Molecular Sciences* **22**, 12926 (2021).